# MAMBA INTEGRATED WITH PHYSICS PRINCIPLES MASTERS LONG-TERM CHAOTIC SYSTEM FORECASTING

## ABSTRACT

Long-term forecasting of chaotic systems remains a fundamental challenge due to the intrinsic sensitivity to initial conditions and the complex geometry of strange attractors. Conventional approaches, such as reservoir computing, typically require training data that incorporates long-term continuous dynamical behavior to comprehensively capture system dynamics. While advanced deep sequence models can capture transient dynamics within the training data, they often struggle to maintain predictive stability and dynamical coherence over extended horizons. Here, we propose PhyxMamba, a framework that integrates a Mamba-based state-space model with physics-informed principles to forecast long-term behavior of chaotic systems given short-term historical observations on their state evolution. We first reconstruct the attractor manifold with time-delay embeddings to extract global dynamical features. After that, we introduce a generative training scheme that enables Mamba to replicate the physical process. It is further augmented by multi-patch prediction and attractor geometry regularization for physical constraints, enhancing predictive accuracy and preserving key statistical properties of systems. Extensive experiments on simulated and real-world chaotic systems demonstrate that PhyxMamba delivers superior forecasting accuracy and faithfully captures essential statistics from short-term historical observations.

## 1 INTRODUCTION

Chaotic systems, defined by their deterministic nature and extreme sensitivity to initial conditions, are pervasive across a wide range of scientific and engineering disciplines, including climate science (Shukla, 1998; Rind, 1999), epidemiology (Bashkirtseva et al., 2020; Jones & Strigul, 2021), neuroscience (Jia et al., 2023; Vignesh et al., 2025), and financial modeling (Wang et al., 2021b; 2022; Vogl, 2022). Understanding the dynamical evolution of these systems is critical not only for forecasting future behavior but also for enabling informed decision-making (Kumar & Sharma, 2021; Shen et al., 2023). However, accurate long-term prediction remains inherently challenging due to the chaoticity, where minor initial errors of two trajectories can be exponentially amplified over time (Levy, 1994). Despite the challenges, long-term evolution of these systems is often bounded and confined to a subset of the state space, known as *strange attractors* (Ott, 1981; Grassberger & Procaccia, 1983), which offer a foundation for learning their dynamics.

Existing paradigms for chaotic system forecasting range from empirical models (Lorenz, 1996; 2017; Rössler, 1976; Rossler, 1979) to contemporary data-driven machine learning and deep learning approaches (Hu et al., 2024a). Despite notable progress, these approaches face inherent limitations in long-term forecasting tasks. Empirical models struggle to generalize to real-world scenarios where system parameters are difficult to determine. Machine learning methods such as reservoir computing (RC) (Pathak et al., 2018; Gauthier et al., 2021; Li et al., 2024) and piecewise-linear recurrent neural network (PLRNN) (Mikhaeil et al., 2022; Hess et al., 2023) require training data that incorporates long-term continuous dynamical behavior to comprehensively capture system dynamics, which is often unavailable in practical applications. Although deep learning-based sequence models (Wu et al., 2021; Zhang & Yan, 2023; Liu et al., 2023a; Wang et al., 2024b;a) have achieved considerable success, they are not specifically tailored to chaotic systems and tend to degrade in performance over extended forecasting horizons.

Consequently, forecasting the long-term dynamics of chaotic systems given short-term historical observations on their states remains a largely underexplored scientific problem in existing research. Effectively addressing this problem necessitates overcoming three fundamental and interrelated challenges: (i) ***Extracting complex dynamical information from short observation windows.*** It involves effectively inferring global system dynamics with short, localized observational data that offers an incomplete view of the system's full dynamical pattern. (ii) ***Maintaining long-term predictive stability and dynamical coherence.*** It requires models to autonomously generate dynamically consistent trajectories over extended future horizons, which involves learning the system's underlying generative process rather than only capturing transient patterns from historical data. (iii) ***Reproducing key statistical properties and attractor geometry.*** It demands that generated long-term forecasts should faithfully reproduce the system's statistical properties and maintain the geometric integrity of its strange attractor, preventing degeneration into spurious attractors or collapse into trivial states.

Recently, state-space models (SSMs) (Hamilton, 1994; Rangapuram et al., 2018), with their strong theoretical foundations in dynamical systems and capacity to represent latent system states over time, provide a natural and powerful framework for modeling such complex temporal behaviors. Therefore, to address the above challenges, we design PhyxMamba, a framework that unifies a Mamba-based state-space model (Gu & Dao, 2023; Dao & Gu, 2024) with physics-informed principles to capture the underlying dynamics of chaotic systems. Specifically, we first employ a physics-informed representation that integrates global and inter-variable dynamics information via time-delay embedding, equipping the model with a comprehensive view of the attractor manifold to enhance the learning of the physical evolution process. We then design a generative next-token training approach for the Mamba backbone to capture the underlying dynamics of the system. It further integrates multi-token prediction during training to enforce physically consistent long-term evolution. After that, we implement an additional student-forcing training stage, refining the learning of the physical process to mitigate error accumulation during autoregressive forecasting. In this stage, we apply Maximum Mean Discrepancy (MMD) (Li et al., 2017; Wang et al., 2021a) to regularize the model, enforcing statistical consistency between the generated trajectories and the true system dynamics. Our contributions can be summarized as follows:

- **Robust long-term forecasting of chaotic systems from short-term historical observations.** We tackle the challenging and underexplored problem of achieving reliable long-term predictions in chaotic systems given solely short-duration historical contexts, a scenario where traditional methods often fall short.

- **A physics-informed Mamba-based forecasting framework.** We propose a novel framework that synergistically combines physics-informed embedding based on chaotic system theory with a generative Mamba state-space model, alongside a tailored regularization strategy to ensure both high predictive accuracy and faithful preservation of key statistical properties.

- **Comprehensive empirical validation on diverse chaotic systems.** We conduct extensive experiments on both simulated and real-world chaotic datasets, demonstrating that our proposed PhyxMamba framework significantly outperforms existing baselines in long-term forecasting, while capturing essential statistical properties and attractor features from limited observation windows.

## 2 PRELIMINARIES AND PROBLEM FORMULATION

In this section, we first formalize the problem of long-term forecasting for chaotic systems using short-horizon observational data. Then, we introduce the fundamental concepts and notations essential for understanding chaotic system forecasting.

**Chaotic Dynamics.** A chaotic system is defined by a deterministic dynamical system exhibiting sensitive dependence on initial conditions, leading to exponentially diverging trajectories in phase space despite being governed by deterministic rules. Formally, consider a continuous-time dynamical system described by the ordinary differential equation (ODE):

$$\frac{d\boldsymbol{x}(t)}{dt} = \boldsymbol{f}(\boldsymbol{x}(t)), \tag{1}$$

where $\boldsymbol{x}(t) \in \mathbb{R}^V$ is the system state at time $t$, $V$ denotes the dimensionality of the system, and $\boldsymbol{f}(\cdot)$ is a nonlinear vector field governing the dynamics. A hallmark of chaotic systems is the presence of a *strange attractor*, a compact set $\mathcal{A} \in \mathbb{R}^V$ in phase space to which trajectories converge

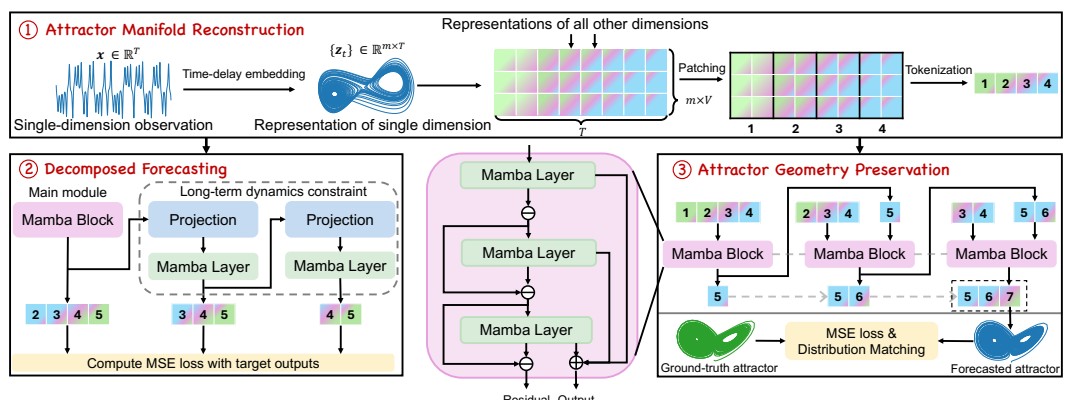

Figure 1: Overview of the proposed framework PhyxMamba.

over time, exhibiting complex, non-periodic behavior. The strange attractor is characterized by its fractal geometry (Theiler, 1990; Fernández-Martínez & Sánchez-Granero, 2014) and invariant measures (Jiang et al., 2023; Schiff et al., 2024), which describe the long-term system statistical properties.

**Lyapunov Exponent and Lyapunov Time.** Lyapunov exponents represent a set of quantities that describe the exponential rate of separation of initially proximate trajectories in a dynamical system. They serve as critical indicators that measure the chaotic behavior of the system. Formally, given the initial separation vector $\boldsymbol{\delta}_0$ of two trajectories, their divergence is given by:

$$|\boldsymbol{\delta}(t)| \approx e^{\lambda t}|\boldsymbol{\delta}_0|, \tag{2}$$

where $\lambda$ denotes the Lyapunov exponent. Given the directional dependence of the separation vector, the Lyapunov exponent exhibits variable values. Among these, we focus on the maximal Lyapunov exponent (MLE), which serves as a key indicator of the system's predictability. Typically, a positive MLE indicates that the system is chaotic. The maximal Lyapunov exponent is defined as follows:

$$\lambda_{\max} = \lim_{t \to \infty} \lim_{|\boldsymbol{\delta}_0| \to 0} \frac{1}{t} \ln \frac{|\boldsymbol{\delta}(t)|}{|\boldsymbol{\delta}_0|}, \tag{3}$$

where $|\boldsymbol{\delta}_0| \to 0$ guarantees the applicability of the linear approximation across all times (Vulpiani, 2010). Lyapunov time is defined as the reciprocal of MLE, i.e., $T_L = 1/\lambda_{\max}$, and represents the characteristic timescale over which a dynamical system exhibits chaotic behavior.

**Problem Formulation.** Given a historical observation window of $T$ time steps, denoted as $\{\boldsymbol{x}(t)\}_{t=1}^{T}$, we aim to effectively forecast the system's future states $\{\hat{\boldsymbol{x}}(t)\}_{t=T+1}^{T+H}$ for as long a horizon $H$ as possible, where $T \ll H$. The forecasting task involves two key objectives: (1) achieving accurate point-wise forecasting, i.e., $\hat{\boldsymbol{x}}(t) \approx \boldsymbol{x}(t), \forall t \in [T+1, T+H]$, (2) preserving the intrinsic dynamical properties, such as the geometry of the strange attractor and key statistical invariants, of the system. Due to the sensitivity of chaotic systems, the latter becomes increasingly important as precise state forecasting grows difficult beyond multiple Lyapunov times. The specific evaluation metrics are detailed in Section 4.1 and Appendix C.2.

## 3 METHODS

In this section, we propose an effective method for capturing the dynamics of underlying chaotic systems. We illustrate the overall pipeline in Figure 1. Here we provide detailed descriptions of all procedures, including attractor manifold reconstruction via time-delay embedding, decomposed forecasting with generative Mamba, and attractor geometry preservation via distribution matching.

### 3.1 ATTRACTOR MANIFOLD RECONSTRUCTION VIA TIME-DELAY EMBEDDING

**Representation.** Prior works (Gilpin, 2023) have established that chaotic systems exhibit increasingly linear behavior when embedded in higher-dimensional representations. Rather than relying on neural

network-based autoencoders for representation learning, we propose a physics-informed pipeline to construct high-dimensional representations for chaotic systems. Takens' embedding theorem (Takens, 2006) demonstrates that we can reconstruct the manifold of the high-dimensional system attractor through time-delay embedding on a single dimension of the observation series of this system (Hu et al., 2024a;b; Wu et al., 2024a). Specifically, given a univariate trajectory $\boldsymbol{x} \in \mathbb{R}^T$ from a chaotic system's attractor with $T$ observed steps, we define two hyperparameters: the embedding dimension $m$ and the time delay $\tau$, and the high-dimensional trajectory is then constructed as follows:

$$\boldsymbol{z}_t = (x_{t-(m-1)\tau}, x_{t-(m-2)\tau}, \cdots, x_t) \in \mathbb{R}^m, \tag{4}$$

where $m$ and $\tau$ denote the embedding dimension and the delayed time step, respectively, and we choose them using CC methods (Kim et al., 1999; Hu et al., 2024b) (Appendix B.1). Consequently, $\{\boldsymbol{z}_t\}$ of different time steps $t$ form an attractor within $m$-dimensional space. It can reconstruct a homeomorphic topological structure with the original attractor if $m$ exceeds twice the system's intrinsic dimension (Vlachos & Kugiumtzis, 2008). To ensure that the reconstructed attractor maintains the same temporal length as $\boldsymbol{x}$ for seamless downstream processing, we apply zero-padding to $\boldsymbol{x}$ before constructing the representations with Equation (4). Thus, we derive $\mathbb{R}^m$ representation for each variable. We define the combined representations from different variables as $\boldsymbol{Z} \in \mathbb{R}^{V \times T \times m}$, where $V$ represents the system's dimensionality. This physics-informed representation pipeline effectively captures the intrinsic geometric properties of the chaotic attractor, yielding interpretable embeddings.

**Patching and Embedding.** To investigate the dynamics of chaotic systems, it is essential to capture temporal correlations across time steps in observed data. However, individual time steps often convey limited temporal semantic information. Inspired by foundational works (Nie et al., 2022; Liu et al., 2024b), we partition the input observation series into patches to encode the local structure of the chaotic system's strange attractor, which enables modeling of the system's global behavior through inter-patch features. Formally, given the physics-informed representation $\boldsymbol{Z} \in \mathbb{R}^{V \times T \times m}$, we segment it into $N = \lfloor \frac{T}{D} \rfloor$ non-overlapping patches $\boldsymbol{P} \in \mathbb{R}^{N \times (D*V*m)}$ with patch size $D$. After that, we map each patch $\boldsymbol{P}_i$ to a $d$-dimensional latent embedding $\boldsymbol{S}_i$ through linear mapping, formulated as:

$$\boldsymbol{S}_i = \boldsymbol{W}_{\text{emb}} \boldsymbol{P}_i + \boldsymbol{b}_{\text{emb}}, \tag{5}$$

where $\boldsymbol{W}_{\text{emb}}$ and $\boldsymbol{b}_{\text{emb}}$ are learnable parameters. Patching and embedding process facilitate the extraction of localized temporal patterns. It also reduces the computational complexity associated with processing time series with multiple steps.

## 3.2 Decomposed Forecasting with Generative Mamba

Achieving long-term system forecasting requires the model to understand the underlying system dynamics. To this end, such a model should be generative within the dynamical system contexts (Brenner et al., 2024), aiming to produce forecasts that preserve the long-term statistical properties of the observed system. In contrast, most conventional time series forecasting models (Wu et al., 2021; Zhang & Yan, 2023; Liu et al., 2023a) typically perform sequence-to-sequence prediction, which may not align with the autoregressive nature of dynamical systems and often lack mechanisms to ensure long-term coherence. Inspired by the recent success of decoder-only large foundation models (Brown et al., 2020; Liu et al., 2024b), we employ a generative training strategy, enabling the model to predict the next patch autoregressively. The prediction for patch $\boldsymbol{P}_i$ can be formulated as follows:

$$\hat{\boldsymbol{P}}_i = f(\boldsymbol{P}_0, \boldsymbol{P}_1, \cdots, \boldsymbol{P}_{i-1}) \tag{6}$$

We implement Equation (6 with a stack of Mamba layers. Instead of naively stacking these layers, we adopt a residual stacking (Oreshkin et al., 2020; Challu et al., 2023) architecture designed to decompose the underlying system dynamics into distinct components, with each Mamba layer learning to model different aspects of the dynamics. It reflects the inherent composite nature of complex systems. Specifically, for the $l$-th Mamba layer, we have:

$$\mathbf{h}_i^{(l)} = \text{Mamba}^{(l)}(\mathbf{r}_{0:i}^{(l-1)}), \quad \widehat{\boldsymbol{E}}_i^{(l)} = \text{Decoder}_e(\mathbf{h}_i^{(l)}), \quad \mathbf{r}_i^{(l)} = \mathbf{r}_i^{(l-1)} - \hat{\boldsymbol{E}}_i^{(l)} \tag{7}$$

where $\mathbf{r}_i^{(l)}$ denotes the residual at the $l$-th layer and $\mathbf{r}_i^{(0)} = \boldsymbol{S}_i$. The decoder is implemented using linear layers, designed to efficiently map the learned representations to the desired output space with minimal computational overhead. The predicted next patch $\hat{\boldsymbol{P}}_{i+1}$ is obtained from

$\widehat{\boldsymbol{P}}_{i+1} = \text{Decoder}_p(\sum_{l=1}^{L} \widehat{\boldsymbol{E}}_i^{(l)})$, where $L$ denotes the number of Mamba layers in the residual stacking architecture, and the decoder is likewise composed of linear layers.

Naive next-patch prediction strategy risks primarily capturing local dynamics, which may hinder the model's ability to maintain long-term predictive stability. To mitigate this, we impose a learning constraint through a multi-patch prediction (MPP) objective, encouraging the model to capture the global dynamics of the underlying system, which has demonstrated effectiveness in large language models (Liu et al., 2024a; Guo et al., 2025). It enforces the model to consider long-term information while predicting future patches. Specifically, we employ $M$ auxiliary modules to predict $M$ subsequent patches. Each MPP module consists of a dedicated Mamba layer, and a fusion projection layer $\psi(\cdot)$. At the prediction depth $m$ for patch $i$, the module integrates the hidden representation $\mathbf{v}_i^{m-1} \in \mathbb{R}^d$ from the previous prediction depth $m-1$ with the embedding of the $(i+m)$-th patch, $\boldsymbol{S}_{i+m} \in \mathbb{R}^d$, through the fusion projection layer, formally expressed as follows:

$$\mathbf{v}_i^{m'} = \psi([\text{RMSNorm}(\mathbf{v}_i^{m-1}), \boldsymbol{S}_{i+m}]), \tag{8}$$

where we employ RMSNorm (Zhang & Sennrich, 2019) layers for normalization, $[\cdot; \cdot]$ denotes the concatenation operation, and $\mathbf{v}_i^1 = \sum_{l=1}^{L} \hat{\boldsymbol{E}}_{i-1}^{(l)}$. The fusion embedding $\mathbf{v}_i^{m'}$ is subsequently passed through the Mamba layer of the corresponding MPP module to produce the updated representation $\mathbf{v}_i^m$. This representation is then used to predict the $m$-th future patch via the module's output prediction head. The process is formalized as follows:

$$\mathbf{v}_{1:N-m}^m = \text{Mamba}(\mathbf{v}_{1:N-m}^{m'}), \quad \hat{\boldsymbol{P}}_{i+m+1}^m = \text{Decoder}_p(\mathbf{v}_i^m). \tag{9}$$

where $N$ represents the length of the input patch sequence. Therefore, the generative training objective of next-patch and multi-patch prediction can be expressed as:

$$\mathcal{L}_{\text{next}} = \frac{1}{NB} \sum_{b=1}^{B} \sum_{i=2}^{N} ||\boldsymbol{P}_i - \hat{\boldsymbol{P}}_i||^2, \quad \mathcal{L}_{\text{MPP}}^m = \frac{1}{(T-1+m)B} \sum_{b=1}^{B} \sum_{i=2+m}^{T} ||\boldsymbol{P}_i - \hat{\boldsymbol{P}}_i^m||, \tag{10}$$

where $B$ represents the number of training samples within a batch, and both $\mathcal{L}_{\text{next}}$ and $\mathcal{L}_{\text{MPP}}^m$ are implemented with the teacher forcing strategy (Mikhaeil et al., 2022; Hess et al., 2023). The overall training objective is $\mathcal{L} = \mathcal{L}_{\text{next}} + \frac{\lambda_p}{M} \sum_{m=1}^{M} \mathcal{L}_{\text{MPP}}^m$, where $\lambda_p$ controls the objective importance. We utilize multi-patch prediction exclusively in the training process to steer the model's representation learning and enhance data efficiency. At inference, the multi-patch prediction modules are *excluded*, and we rely solely on the main Mamba block for autoregressive next-patch prediction during forecasting.

### 3.3 ATTRACTOR GEOMETRY PRESERVATION VIA DISTRIBUTION MATCHING

While the *teacher forcing* strategy promotes training stability, it can cause the model to become overly reliant on ground-truth patches. This reliance creates a fundamental mismatch with the autoregressive nature of inference, where prediction errors can accumulate and ultimately impair performance on long-horizon forecasting tasks.

To mitigate this issue, we introduce an additional *student-forcing* training phase, wherein the model autoregressively generates $W$ patches based on its historical predictions and computes the MSE loss:

$$\mathcal{L}_{\text{stu}} = \frac{1}{WB} \sum_{b=1}^{B} \sum_{j=1}^{W} ||\boldsymbol{P}_j - \hat{\boldsymbol{P}}_j||^2. \tag{11}$$

To complement the standard Mean Squared Error (MSE) loss, which targets point-wise forecasting accuracy, we introduce a regularization term to preserve the statistical properties of the system's dynamics. It is well-established that the trajectories of dissipative chaotic systems are confined to a strange attractor, on which the probability distribution of states forms an invariant measure (Jiang et al., 2023; Schiff et al., 2024; Cheng et al., 2025). To enforce this property, we seek to minimize the discrepancy between the empirical distributions formed by the predicted and observed states. Specifically, within a given batch, let $\{\boldsymbol{u}^{(i)}\}_{i=1}^n \sim p_{\text{hist}}$, $\{\boldsymbol{v}^{(i)}\}_{i=1}^n \sim p_{\text{pred}}$, and $\{\boldsymbol{w}^{(i)}\}_{i=1}^n \sim p_{\text{gt}}$ represent the sets of states from the historical, predicted future, and ground-truth future trajectories, respectively. We employ the Maximum Mean Discrepancy (MMD; see Appendix B.3) to quantify the

distance between empirical distributions $p_{\text{hist}}$, $p_{\text{pred}}$, and $p_{\text{gt}}$ formed by these trajectories. For instance, the MMD between the future ground-truth and predicted state distributions is estimated as:

$$\widehat{\text{MMD}}^2(p_{\text{gt}}, p_{\text{pred}}) = \frac{1}{n^2}\sum_{i,j}\kappa(\boldsymbol{w}^{(i)}, \boldsymbol{w}^{(j)}) + \frac{1}{n^2}\sum_{i,j}\kappa(\boldsymbol{v}^{(i)}, \boldsymbol{v}^{(j)}) - \frac{2}{n^2}\sum_{i,j}\kappa(\boldsymbol{w}^{(i)}, \boldsymbol{v}^{(j)}). \quad (12)$$

Following prior works (Schiff et al., 2024; Li et al., 2015; Seeger, 2004), we implement $\kappa$ as a mixture of rational quadratic kernels (Appendix B.3). The MMD between the historical and predicted state distributions, $\widehat{\text{MMD}}^2(p_{\text{hist}}, p_{\text{pred}})$, is computed analogously by replacing $\{\boldsymbol{w}^{(i)}\}_{i=1}^n$ with $\{\boldsymbol{u}^{(i)}\}_{i=1}^n$. The full regularization term is thus defined as:

$$\mathcal{L}_{\text{reg}} = \widehat{\text{MMD}}^2(p_{\text{hist}}, p_{\text{pred}}) + \lambda_c\widehat{\text{MMD}}^2(p_{\text{gt}}, p_{\text{pred}}), \quad (13)$$

where $\lambda_c$ is a hyperparameter that controls the relative importance of the two components. The first term enforces statistical consistency between the historical dynamics and the model's predictions, while the second aligns the predicted state distribution with that of the ground truth. The total objective for the student forcing phase is then a weighted sum of the primary and regularization losses: $\mathcal{L} = \mathcal{L}_{\text{stu}} + \lambda_r\mathcal{L}_{\text{reg}}$, where $\lambda_r$ is a hyperparameter for the regularization strength.

## 4 EXPERIMENTS

### 4.1 EXPERIMENTAL SETUP

**Datasets.** We evaluate our method on three standard simulated chaotic systems: the 3-dimensional Lorenz63 (Lorenz, 2017), Rössler (Rössler, 1976; Rossler, 1979), and 5-dimensional Lorenz96 (Lorenz, 1996), chosen for their well-characterized attractors and varying complexity (Mikhaeil et al., 2022; Hess et al., 2023). To complement these, we use real-world datasets: the 5-dimensional Electrocardiogram (ECG)(Reiss et al., 2019) and 64-dimensional Electroencephalogram (EEG)(Schalk et al., 2004), both exhibiting chaotic traits evidenced by positive maximum Lyapunov exponents. Full dataset details are provided in Appendix C.1.

**Evaluation Metrics.** To comprehensively assess the performance of our proposed method and competing baselines, we employ a suite of evaluation metrics that capture both point-wise predictive accuracy and long-term statistical fidelity of chaotic systems. The former encompasses metrics such as 1-step Error, symmetric mean absolute percentage error (sMAPE), and valid prediction time (VPT); while the latter is quantified using the correlation dimension error ($D_{\text{frac}}$) and KL divergence between attractors ($D_{\text{stsp}}$). Notably, the timescales for sMAPE and VPT are measured in units of the system's Lyapunov time ($T_L$). Details of evaluation metrics are demonstrated in Appendix C.2.

**Baselines.** Baseline models can be broadly categorized into two primary paradigms: The first category comprises physics-informed dynamical systems models. We incorporate Koopa (Liu et al., 2023b), PLRNN (Mikhaeil et al., 2022), nVAR (Gauthier et al., 2021), PRC (Srinivasan et al., 2022), and HoGRC (Li et al., 2024) for comparison. The second category includes general time series forecasting models. This category includes NBEATS (Oreshkin et al., 2019), NHiTS (Challu et al., 2023), CrossFormer (Zhang & Yan, 2023), PatchTST (Nie et al., 2022), TimesNet (Wu et al., 2022), TiDE (Das et al., 2023), iTransformer (Liu et al., 2023a), DLinear (Toner & Darlow, 2024), NSFormer (Liu et al., 2022), FEDFormer (Zhou et al., 2022), and AutoFormer (Wu et al., 2021). We also investigate zero-shot performance of time series foundation models like Timer (Liu et al., 2024b) and Chronos (Ansari et al., 2024), and demonstrate the results in Appendix E.1.

**Experimental Settings.** Our goal is to predict the long-term dynamics of chaotic systems using short-term observations. Data from the Lorenz63, Rossler, Lorenz96, and ECG systems are resampled according to their Lyapunov time ($T_L$), where each Lyapunov time corresponds to 30 time steps. For the EEG system, data are sampled such that one Lyapunov time covers 60 time steps. During training, our model utilizes data spanning one Lyapunov time ($1T_L$) to perform next-patch prediction with teacher forcing, where the patch length $|\boldsymbol{S}_i| < T_L$. Our model autoregressively predicts the subsequent $T_L$ trajectory based on the preceding $T_L$ segment in the student forcing stage. Similarly, baseline models are trained to predict trajectories of length $T_L$ based on observed data of the same duration $T_L$. For evaluation, we measure test performance by providing models with observation trajectories of length $T_L$ and tasking them with predicting the subsequent trajectories spanning $10T_L$. We then compute and report all relevant metrics accordingly.

Table 1: Overall performances on simulated and real-world datasets. The best performance of each metric is marked in **bold**, and the second-best performance is underlined.

| System | Metric | Ours | NBEATS | NHiTS | CrossFormer | PatchTST | TimesNet | TiDE | iTransformer | Koopa | PLRNN | nVAR | PRC | HoGRC |
|---|---|---|---|---|---|---|---|---|---|---|---|---|---|---|
| Lorenz63 (3D) | VPT (↑) | **5.06** | 3.66 | 2.86 | 1.95 | 0.46 | 1.01 | 0.32 | 0.30 | 0.33 | 0.71 | 0.20 | 1.59 | 1.12 |
| | 1-step Error (↓) | 0.080 | 0.056 | 0.252 | 0.605 | 1.299 | 0.389 | 0.711 | 1.271 | 2.600 | 0.180 | 0.402 | 0.0004 | **0.0002** |
| | sMAPE@1 (↓) | **3.26** | 5.34 | 13.93 | 13.62 | 64.39 | 35.84 | 84.88 | 67.84 | 60.68 | 59.26 | 94.32 | 18.34 | 18.82 |
| | sMAPE@4 (↓) | **22.05** | 37.08 | 52.36 | 64.40 | 94.39 | 87.45 | 97.85 | 95.95 | 90.45 | 95.37 | 107.01 | 124.63 | 93.32 |
| | sMAPE@10 (↓) | **67.29** | 78.46 | 84.18 | 92.96 | 104.24 | 103.25 | 102.83 | 107.96 | 100.54 | 104.43 | 110.23 | 135.42 | 102.96 |
| | $D_{\text{frac}}$ (↓) | **0.060** | 0.075 | 0.067 | 0.090 | 0.227 | 0.303 | 0.696 | 0.638 | 0.333 | 0.270 | 0.792 | 0.578 | 0.089 |
| | $D_{\text{stsp}}$ (↓) | **1.133** | 1.586 | 1.648 | 1.591 | 61.646 | 32.343 | 92.375 | 62.434 | 58.768 | 12.970 | 61.331 | 29.389 | 16.761 |
| Rossler (3D) | VPT (↑) | **9.83** | 6.48 | 0.44 | 0.95 | 0.04 | 0.26 | 0.04 | 0.07 | 0.05 | 0.44 | 0.01 | 1.13 | 0.83 |
| | 1-step Error (↓) | 0.007 | 0.017 | 0.024 | 0.092 | 0.422 | 0.102 | 0.382 | 0.659 | 0.050 | 0.288 | | **0.0009** | 0.0036 |
| | sMAPE@1 (↓) | **2.84** | 14.88 | 39.90 | 27.11 | 89.16 | 42.48 | 98.48 | 85.85 | 67.78 | 51.36 | 94.95 | 28.38 | 31.03 |
| | sMAPE@4 (↓) | **3.55** | 22.68 | 64.07 | 38.32 | 111.19 | 107.31 | 116.59 | 121.07 | 99.61 | 84.09 | 115.42 | 128.40 | 111.01 |
| | sMAPE@10 (↓) | **12.64** | 41.22 | 83.54 | 62.43 | 129.67 | 138.19 | 126.01 | 128.48 | 122.36 | 107.07 | 140.49 | 130.26 | 115.82 |
| | $D_{\text{frac}}$ (↓) | **0.049** | 0.069 | 0.088 | 0.074 | 0.158 | 0.530 | 0.455 | 0.583 | 0.497 | 0.400 | 0.750 | 0.770 | 0.611 |
| | $D_{\text{stsp}}$ (↓) | **0.034** | 0.137 | 0.538 | 0.187 | 8.423 | 4.985 | 15.034 | 10.807 | 5.721 | 3.950 | 7.391 | 4.549 | 4.068 |
| Lorenz96 (5D) | VPT (↑) | **1.66** | 0.92 | 0.61 | 0.28 | 0.05 | 0.06 | 0.06 | 0.02 | 0.00 | 0.23 | 0.17 | 0.99 | 0.68 |
| | 1-step Error (↓) | 0.135 | 0.100 | **0.071** | 0.583 | 0.948 | 1.020 | 0.480 | 1.936 | 2.800 | 0.230 | 0.519 | 0.056 | 0.087 |
| | sMAPE@1 (↓) | **20.28** | 37.51 | 51.37 | 56.64 | 116.63 | 95.93 | 123.41 | 126.08 | 114.74 | 89.90 | 86.04 | 36.42 | 47.75 |
| | sMAPE@4 (↓) | **81.18** | 100.46 | 107.53 | 111.21 | 130.39 | 122.15 | 128.61 | 128.10 | 127.03 | 124.11 | 123.77 | 102.44 | 106.73 |
| | sMAPE@10 (↓) | **113.56** | 121.77 | 125.51 | 124.31 | 134.08 | 127.19 | 130.62 | 129.23 | 130.06 | 132.05 | 120.80 | 116.02 | 120.34 |
| | $D_{\text{frac}}$ (↓) | **0.154** | 0.172 | 0.195 | 0.196 | 1.262 | 1.172 | 1.273 | 1.252 | 0.806 | 0.370 | 0.283 | 0.179 | **0.154** |
| | $D_{\text{stsp}}$ (↓) | **15.440** | 18.043 | 18.132 | 36.191 | 141.306 | 102.414 | 174.493 | 170.792 | 128.611 | 24.020 | 23.537 | 16.834 | 17.287 |
| ECG (5D) | VPT (↑) | **0.93** | 0.61 | 0.59 | 0.33 | 0.29 | 0.32 | 0.01 | 0.51 | 0.21 | 0.20 | 0.04 | 0.06 | 0.19 |
| | 1-step Error (↓) | **0.017** | 0.024 | 0.024 | 0.070 | 0.072 | 0.083 | 0.316 | 0.063 | 0.087 | 0.032 | 0.092 | 0.398 | **0.017** |
| | sMAPE@1 (↓) | **37.14** | 50.77 | 50.64 | 58.70 | 64.83 | 60.84 | 136.53 | 55.59 | 60.63 | 116.00 | 147.87 | 136.60 | 144.10 |
| | sMAPE@4 (↓) | **73.64** | 87.67 | 84.19 | 101.15 | 124.47 | 120.39 | 130.87 | 117.09 | 122.57 | 145.26 | 183.33 | 139.58 | 185.79 |
| | sMAPE@10 (↓) | **99.68** | 108.20 | 112.03 | 124.75 | 156.24 | 132.97 | 128.69 | 147.44 | 151.54 | 152.53 | 191.31 | 145.67 | 195.76 |
| | $D_{\text{frac}}$ (↓) | **0.036** | 0.038 | 0.078 | 0.443 | 0.218 | 0.143 | 0.219 | 0.242 | 0.245 | 0.529 | 0.642 | 0.147 | 0.184 |
| | $D_{\text{stsp}}$ (↓) | **0.004** | 0.034 | 0.035 | 0.156 | 0.440 | 0.682 | 0.834 | 0.439 | 0.463 | 1.592 | 0.638 | 3.265 | 4.015 |
| EEG (64D) | VPT (↑) | **9.43** | 0.083 | 0.253 | 0.023 | 0.051 | 0.00 | 0.036 | 0.015 | 0.018 | 0.029 | 0.04 | 0.035 | 0.047 |
| | 1-step Error (↓) | **0.031** | 0.056 | 0.033 | 0.169 | 0.068 | 0.562 | 0.080 | 0.201 | 0.257 | 0.087 | 1.063 | 0.059 | **0.031** |
| | sMAPE@1 (↓) | **14.21** | 79.28 | 57.90 | 122.58 | 112.98 | 126.32 | 116.12 | 120.88 | 125.12 | 147.84 | 145.95 | 119.85 | 181.02 |
| | sMAPE@4 (↓) | **18.12** | 119.11 | 98.01 | 144.06 | 139.23 | 144.06 | 139.01 | 142.16 | 139.78 | 168.68 | 148.27 | 138.43 | 194.70 |
| | sMAPE@10 (↓) | **21.87** | 131.69 | 126.07 | 154.63 | 145.33 | 145.86 | 144.80 | 146.10 | 143.83 | nan | 149.11 | 144.52 | 198.97 |
| | $D_{\text{frac}}$ (↓) | **0.180** | 0.620 | 0.961 | 0.516 | 1.295 | 1.265 | 1.314 | 1.195 | 1.295 | 1.675 | 0.463 | 1.323 | 1.282 |
| | $D_{\text{stsp}}$ (↓) | **0.259** | 11.982 | 12.000 | 24.818 | 33.946 | 18.140 | 30.614 | 30.080 | 35.596 | 25.688 | 9.832 | 33.379 | 37.355 |

## 4.2 OVERALL PERFORMANCE

The empirical performance of our model is validated on both simulated and real-world datasets, with quantitative results presented in Table 1 and Appendix E.1, respectively. Our findings demonstrate the following key advantages: (1) ***Our model consistently achieves state-of-the-art point-wise forecasting accuracy across all evaluated datasets.*** Specifically, our model sustains high point-wise accuracy over extended forecasting horizons. For instance, our model attains a VPT of $9.83T_L$ and $9.43T_L$ on the Rossler simulation and the real-world EEG dataset, respectively. These VPT values significantly surpass those of the best-performing baselines. This level of predictive power is achieved using an observation length of merely one Lyapunov time ($T_L$), while models like PRC and HoGRC typically require long trajectory data to capture the full system dynamics (Li et al., 2024). Moreover, it outperforms all baseline methods regarding sMAPE at 1, 4, and 10 Lyapunov times ($T_L$), underscoring our model's proficiency in capturing the system's underlying dynamics. (2) ***Beyond point-wise prediction, our model excels at preserving the long-term statistical characteristics and geometric integrity of the strange attractors inherent in the dynamical systems.*** On the simulated Rossler dataset, for example, our model yields $D_{\text{frac}}$ of 0.049 and $D_{\text{stsp}}$ of 0.034, markedly lower than the best baseline values of 0.069 and 0.137, respectively. Similar results are also observed on the real-world ECG and EEG datasets. Visualizations of the forecasted trajectories provided in Figure 2 further corroborate these quantitative results, illustrating our method's superior ability to reconstruct the attractor's complex geometry. (3) ***Strong short-term predictive accuracy does not necessarily guarantee robust long-term forecasting performance.*** Models exhibiting lower 1-step prediction errors, such as HoGRC, PRC, and NHiTS, often show rapid degradation in their forecasts, quickly diverging into inaccurate prediction regimes. In contrast, our model maintains its predictive fidelity over much longer horizons. It highlights that our model moves beyond superficial correlation capture that may suffice for short-term predictions but fails in the long run.

## 4.3 ABLATION STUDY

To rigorously evaluate the contribution of each design element to our model's overall efficacy, we conducted a comprehensive ablation study. The results of these experiments are detailed in Table 2. We find that removing physics-informed representation results in the most substantial degradation of the attractor's long-term geometric integrity, underscoring its critical role in capturing essential underlying system dynamics. Moreover, ablation of student-forcing led to the most acute decline in point-wise accuracy. It indicates the importance of this strategy in enabling the model to adapt to its

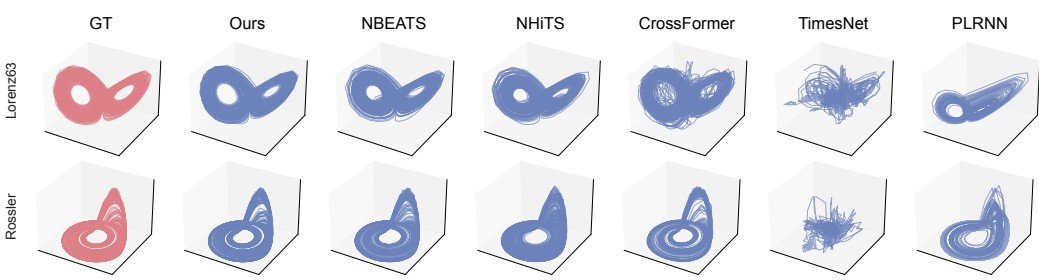

Figure 2: Visualization of forecasting results on Lorenz63 and Rossler systems.

prediction noise during training, which is vital for achieving robust forecasting performance with extended horizons. In summary, these ablation experiments affirm that all architectural designs and training strategies are crucial for achieving the superior performance demonstrated by our proposed method. We also benchmark our full model against an encoder-oriented Mamba variant. Our model excels by comprehensively capturing the underlying evolutionary dynamics of the system, thereby achieving more robust and accurate long-range predictions, whereas encoder-oriented Mamba is limited to learning direct predictive mappings.

Table 2: Model performances when removing each of our designs. The best performance of each metric is marked in **bold**, and the second-best performance is underlined. (PIR: physics-informed representation; RS: residual stacking; MPP: multi-patch prediction; SF: student forcing; MMD: statistical property regularization.)

| System | Rossler (3D) | | | | | | | EEG (64D) | | | | | | |
|---|---|---|---|---|---|---|---|---|---|---|---|---|---|---|
| Metric / Ablation | VPT ($\uparrow$) | 1-step Error ($\downarrow$) | sMAPE@1 / 4 / 10 ($\downarrow$) | | | $D_{\text{frac}}$ ($\downarrow$) | $D_{\text{stsp}}$ ($\downarrow$) | VPT ($\uparrow$) | 1-step Error ($\downarrow$) | sMAPE@1 / 4 / 10 ($\downarrow$) | | | $D_{\text{frac}}$ ($\downarrow$) | $D_{\text{stsp}}$ ($\downarrow$) |
| **Full** | **9.83** | **0.007** | **2.84** | **3.55** | **12.64** | **0.050** | **0.034** | **9.43** | **0.031** | **14.21** | **18.12** | **21.87** | 0.180 | **0.259** |
| w/o PIR | 2.48 | 0.087 | 17.91 | 38.31 | 56.43 | 0.105 | 0.239 | 0.036 | 0.144 | 74.79 | 109.16 | 129.81 | 0.646 | 12.410 |
| w/o RS | 5.51 | 0.055 | 11.29 | 25.07 | 46.55 | 0.075 | 0.150 | 0.044 | 0.161 | 42.69 | 47.40 | 54.37 | **0.143** | 0.927 |
| w/o MPP | 5.66 | 0.063 | 12.48 | 22.99 | 41.47 | 0.091 | 0.193 | 0.045 | 0.114 | 48.13 | 57.18 | 78.87 | 0.424 | 2.220 |
| w/o SF | 1.29 | 0.071 | 27.22 | 37.18 | 55.86 | 0.108 | 0.230 | 2.31 | 0.041 | 24.34 | 33.17 | 42.87 | 0.278 | 0.263 |
| w/o MMD | 7.40 | 0.031 | 8.34 | 17.23 | 37.32 | 0.083 | 0.136 | 4.81 | 0.056 | 23.22 | 27.71 | 31.48 | 0.230 | 0.479 |
| Encoder-oriented | 2.61 | 0.054 | 20.68 | 35.14 | 56.12 | 0.081 | 0.160 | 0.017 | 0.083 | 120.90 | 121.31 | 150.46 | 0.749 | 23.629 |

## 4.4 ROBUST ANALYSIS

**Robustness against Noise.** We begin by evaluating the robustness of our model under varying levels of observational noise. Specifically, we inject Gaussian noise with zero mean and standard deviation $\sigma$, to the trajectories. into the trajectory data. The resulting performance metrics for sMAPE@1 and $D_{\text{stsp}}$ are illustrated in Figure 3(a–d). Our results reveal that although point-wise accuracy (sMAPE@1) deteriorates notably when $\sigma > 0.1$, the long-term consistency metric ($D_{\text{stsp}}$) remains remarkably stable. Given the inherently chaotic nature of the system, maintaining high point-wise accuracy in the presence of random noise is intrinsically challenging. Nonetheless, the sustained stability of the long-term consistency metrics highlights the model's ability to faithfully preserve the geometric structure and essential statistical invariants of the system's strange attractor—a critical property for practical, real-world applications. More results are shown in Appendix E.3.

**Robustness against Training Data Ratio.** We also investigate the effect of the number of training data on the model's performance, and demonstrate the results in Figure 3(e-h). Our findings demonstrate that our approach exhibits significantly higher data efficiency than the best baseline methods. For instance, on the Rossler system, our model requires only 20% of the full training dataset to outperform the best baseline in both point-wise accuracy and long-term geometric consistency. A similar trend is observed on the EEG system, where only 40% of the training data suffices to surpass baseline performance. More results are shown in Appendix E.3.

## 5 RELATED WORKS

**Chaotic System Forecasting.** Chaotic system forecasting has gained significant attention due to its importance across diverse scientific and engineering fields. A key breakthrough is reservoir computing

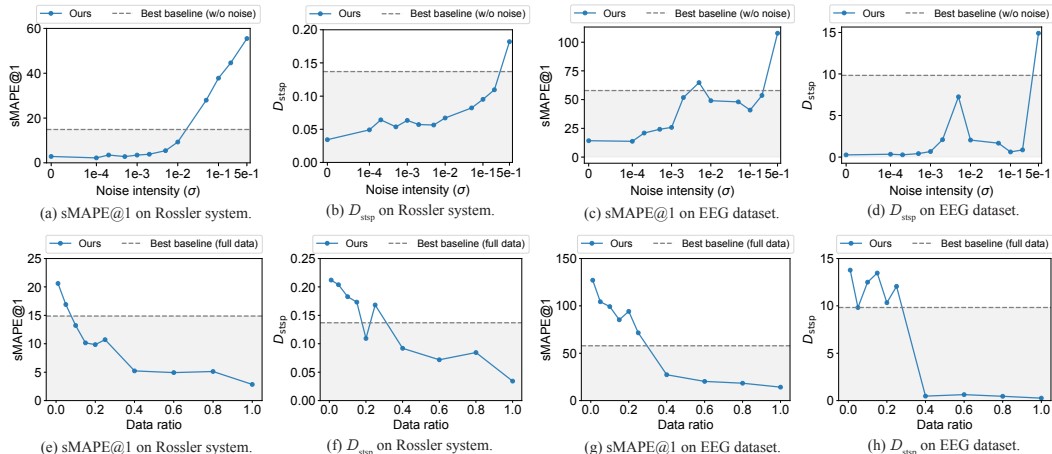

Figure 3: Model robustness against noise and training data on Rossler system and EEG dataset.

(RC) (Gauthier et al., 2021; Srinivasan et al., 2022; Li et al., 2024), which uses a fixed, randomly initialized recurrent neural network (RNN) to map inputs into a high-dimensional dynamical space, training only the readout layer for efficient learning. More recently, deep learning models, particularly advanced RNNs like Long Short-Term Memory (LSTM) networks(Hochreiter & Schmidhuber, 1997), have outperformed traditional methods and RC in complex scenarios by capturing temporal dependencies. To mitigate gradient explosion during chaotic trajectory modeling, recent works (Hess et al., 2023; Mikhaeil et al., 2022) employ teacher forcing, substituting internal states with ground truth during training. Additionally, novel physics-informed regularizations based on optimal transport and Maximum Mean Discrepancy (MMD) with neural operators (Jiang et al., 2023; Schiff et al., 2024) help preserve chaotic attractors' invariant measures. Although many time series architectures exist (Nie et al., 2022; Wu et al., 2021; Liu et al., 2023a; Zhang & Yan, 2023; Wang et al., 2024b;a), their performance remains limited for chaotic forecasting. In contrast, our method is explicitly tailored to this challenge, enabling accurate long-term predictions from short-term observations.

**Mamba in Time Series Modeling.** Mamba (Gu & Dao, 2023; Dao & Gu, 2024), rooted in state space models, has recently gained prominence in sequence modeling across fields like natural language processing and computer vision. In time series analysis, Mamba architectures increasingly capture temporal and channel-wise correlations effectively. For example, TimeMachine (Ahamed & Cheng, 2024) and DTMamba (Wu et al., 2024b) employ parallel Mamba modules at low and high resolutions to extract local and global contexts, while S-Mamba (Wang et al., 2025) uses bi-directional layers to model inter-channel dependencies. Other works combine Mamba with complementary modules: Mambaformer (Xu et al., 2024) integrates self-attention for short-term interactions alongside Mamba's long-term modeling, and Bi-Mamba+ (Liang et al., 2024) introduces a forget gate to enhance long-term information retention. However, these methods primarily use Mamba as an encoder for correlation modeling, without explicitly learning intrinsic dynamical processes. Our approach differs by embedding Mamba blocks in a generative framework, achieving both competitive forecasting and effective capture of the system's intrinsic dynamics.

# 6 CONCLUSIONS

In this work, we introduce PhyxMamba, a novel framework that addresses the critical challenge of long-term forecasting of chaotic systems given short-term historical observations. By synergistically integrating a Mamba-based state-space model with physics-informed embedding, our approach effectively achieves point-wise state forecasting accuracy and reproduces key statistical properties of systems. Through extensive experiments on both simulation and real-world chaotic system datasets, we demonstrate the effectiveness in chaotic system forecasting of our model design and superior performance compared with baseline models. Our work paves the way for future research in modeling complex dynamical systems under observation-scarce conditions, with potential applications in climate science, neuroscience, epidemiology, and beyond.

## ETHICS STATEMENT

The authors have read and adhered to the ICLR Code of Ethics. This work is committed to the principles of responsible and ethical research. This research is based on established scientific principles. All datasets, whether simulated or real-world, and prior works that form the foundation of our study are appropriately cited throughout the manuscript. The simulated data (Lorenz63, Rossler, Lorenz96) were generated using standard, publicly available libraries, ensuring reproducibility. This study utilizes two real-world datasets that contain human physiological data: the Electrocardiogram (ECG) dataset and the Electroencephalogram (EEG) dataset. Both datasets are publicly available and were obtained from previously published research, which are both cited accordingly. We did not collect any new data from human subjects.

## REPRODUCIBILITY STATEMENT

To ensure the reproducibility of our results, we have made our code, experimental setup, and data processing procedures publicly available. The complete source code for our proposed PhyxMamba model and all experiments is anonymously hosted on GitHub at `https://anonymous.4open.science/r/PhyxMamba-7F7E`. The repository includes instructions for setting up the environment and running the experiments. Detailed descriptions of the model architecture and training procedures, such as attractor manifold reconstruction and distribution matching, are provided in Section 3 of the main paper. Hyperparameter settings for all experiments on both simulated and real-world datasets are fully documented in Table 3 of Appendix C.3. Furthermore, a comprehensive description of the datasets used, including their sources, generation or processing steps, and splits for training, validation, and testing, can be found in Appendix C.1. The specific evaluation metrics are detailed in Section 4.1 and Appendix C.2.

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

## A  EXTENDED DEMONSTRATION ON MOTIVATION

In this section, we provide details on the motivation for our work. As illustrated in Figure 4, we evaluate the performance of models in forecasting the Lorenz63 system using an observation length equivalent to one Lyapunov time ($1T_L$, Section 4.2 of the main text). We observe that even some recent methods, such as NHiTS, CrossFormer, and HoGRC, fail to achieve satisfactory performance, with the best validation prediction length approaching four Lyapunov times. In contrast, our method can predict for as long as five Lyapunov times ($5T_L$), demonstrating its effectiveness.

Figure 5 summarizes the limitations of three major modeling approaches: theoretical models, physics-informed machine learning (ML) models, and general time series forecasting models. Theoretical models often require carefully calibrated parameters derived from observational data; however, these parameters do not easily adapt to new or changing data, limiting the model's flexibility and robustness. Physics-informed ML models, while effective in reconstructing complex systems, rely heavily on training data that captures rich, long-term dynamical patterns—data that is frequently unavailable in real-world scenarios. General-purpose time series forecasting models, such as those based on Transformer architectures, are typically designed for short-term sequence prediction. As a result, they lack deeper physical insights and often fail to generalize effectively over long forecasting horizons, particularly in chaotic systems.

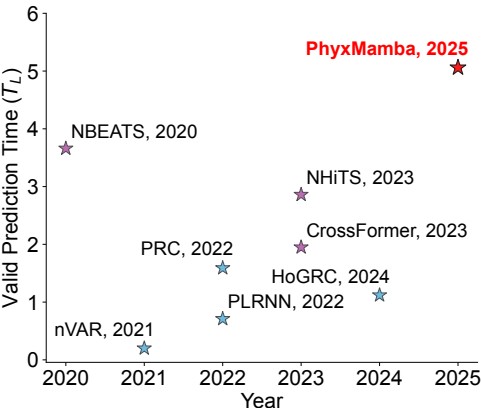

Figure 4: Forecasting performances of representative models on the Lorenz63 system with short-term observations.

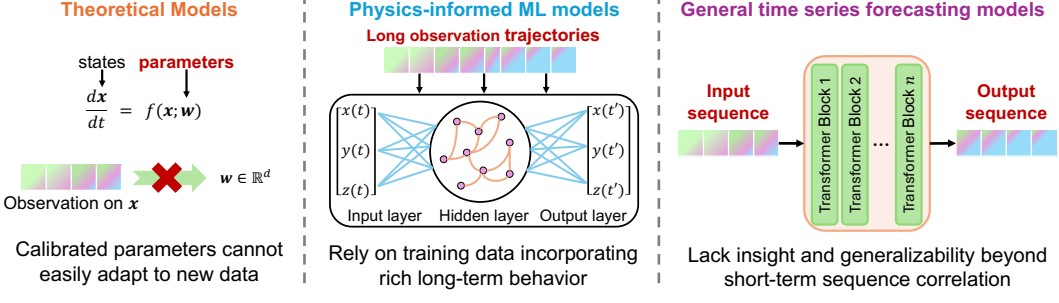

Figure 5: Limitations of existing methods on the chaotic system forecasting.

## B  IMPLEMENTATION DETAILS

In this section, we demonstrate implementation details of our methods.

### B.1 Representation

**Takens' Embedding Theorem.** It is a key result in nonlinear dynamics, which states that the state space of a smooth dynamical system can be reconstructed from time-delay embeddings of a single observed scalar time series. Given a scalar measurement function $h : \mathcal{M} \to \mathbb{R}$ on a compact manifold $\mathcal{M}$ representing the system's state space, the delay embedding map is defined as

$$\Phi(\boldsymbol{x}) = (h(\boldsymbol{x}), h(f(\boldsymbol{x})), h(f^2(\boldsymbol{x})), \dots, h(f^{m-1}(\boldsymbol{x}))), \tag{14}$$

where $f : \mathcal{M} \to \mathcal{M}$ is the system's evolution map and $m$ is the embedding dimension. Takens showed that for a generic choice of $h$, if $m > 2d$ (with $d$ being the dimension of $\mathcal{M}$), the map $\Phi$ is an embedding, meaning it is a diffeomorphism onto its image, thus preserving the topology of the original system.

**Time-delay Embedding.** The most common approach to implement $\Phi(\cdot)$ is the time-delay embedding method. This technique constructs vectors from a single scalar time series as

$$\boldsymbol{y}(t) = (h(\boldsymbol{x}(t)), h(\boldsymbol{x}(t - \tau)), h(\boldsymbol{x}(t - 2\tau)), \dots, h(\boldsymbol{x}(t - (m - 1)\tau))), \tag{15}$$

where $\tau$ is the chosen time delay and $m$ is the embedding dimension. According to Takens' embedding theorem demonstrated above, if $m > 2d$ (with $d$ being the dimension of the original system), the map $\Phi : \boldsymbol{x}(t) \mapsto \boldsymbol{y}(t)$ is a diffeomorphism onto its image, meaning the reconstructed trajectory preserves the essential dynamical properties of the original system.

**CC Methods.** In Section 3.1 of the main text, we present time-delay embedding based on Takens' embedding theorem to derive high-dimensional representations of chaotic systems. The embedding method depends critically on two hyperparameters: the embedding dimension $m$ and the time delay $\tau$. The choice of these parameters profoundly influences the efficacy and accuracy of the representation quality.

Given an observed time series $\boldsymbol{x}$ of a single variable, to select an appropriate time delay $\tau$, we use the Average Mutual Information (AMI), which measures the amount of shared information between $\boldsymbol{x}(t)$ and $\boldsymbol{x}(t + \tau)$. AMI is defined as:

$$\text{AMI}(\tau) = \sum_{x_t, x_{t+\tau}} p(x_t, x_{t+\tau}) \log \frac{p(x_t, x_{t+\tau})}{p(x_t) p(x_{t+\tau})}, \tag{16}$$

where $p(x_t)$ and $p(x_{t+\tau})$ are the marginal probability distributions, and $p(x_t, x_{t+\tau})$ is the joint distribution. The optimal $\tau$ corresponds to the *first local minimum* of $\text{AMI}(\tau)$, ensuring that the embedded components are as independent as possible.

The proper embedding dimension $m$ ensures that trajectories in the reconstructed manifold do not intersect, thereby preserving the topological properties of the original system. To determine the optimal $m$, we employ the False Nearest Neighbors (FNN) method, which quantifies the proportion of points that appear close in a lower-dimensional space but are distant in a higher-dimensional embedding. The FNN measure $F(m)$ is defined as:

$$F(m) = \frac{1}{N - M + 1} \sum_{i=1}^{N-m+1} \log C_i(m, r), \tag{17}$$

where $C_i(m, r)$ denotes the number of points within a radius $r$ of the $i$-th point in the $m$-dimensional space, and $N$ is the length of the time series. Typically, $F(m)$, the false nearest neighbors measure, increases with the embedding dimension $m$ until it reaches a saturation point. The smallest $m$ at which $F(m)$ stabilizes without significant growth should be chosen, as this indicates that higher dimensions do not provide further information about the system's dynamics.

### B.2 State-space Models and Mamba

State-space model (SSM) is a mathematical framework derived from dynamical systems. It describes the relationships between hidden states and observation variables of the system and models their temporal behavior. SSM usually maps continuous inputs $x(t)$ to corresponding outputs $y(t)$ via the state hidden representation $h(t)$:

$$\frac{dh(t)}{dt} = \boldsymbol{A}h(t) + \boldsymbol{B}x(t),$$
$$y(t) = \boldsymbol{C}h(t), \tag{18}$$

where $\boldsymbol{A}, \boldsymbol{B}, \boldsymbol{C}$ represent the parameters of SSMs. Because of the difficulty in acquiring analytical solutions for SSMs, discretization plays an important role in facilitating analysis and solution in the discrete domain. In such a process, the input signal is first sampled at fixed time intervals, and the resulting discrete state-space model can be written as follows:

$$
\begin{aligned}
h_k &= \overline{\boldsymbol{A}} h_{k-1} + \overline{\boldsymbol{B}} x_k, \\
y_k &= \boldsymbol{C} h_k,
\end{aligned}
\tag{19}
$$

where $\overline{\boldsymbol{A}} = f_A(\Delta, \boldsymbol{A})$ and $\overline{\boldsymbol{B}} = f_B(\Delta, \boldsymbol{B})$. Discrete rules $(f_A, f_B)$ are usually chosen as the zero-order hold (ZOH) defined as:

$$
\overline{\boldsymbol{A}} = \exp(\Delta \boldsymbol{A}), \qquad \overline{\boldsymbol{B}} = (\Delta \boldsymbol{A})^{-1}(\exp(\Delta \boldsymbol{A}) - \boldsymbol{I}) \cdot \Delta \boldsymbol{B},
\tag{20}
$$

where $\boldsymbol{\Delta}$ denotes the step size, and $\boldsymbol{I}$ is the identity matrix.

Traditional SSMs are implemented as time-invariant systems, where transfer parameters (e.g., $\boldsymbol{B}$ and $\boldsymbol{C}$) are identical at all time steps. This linear time-invariant (LTI) characteristic, while enabling efficient computation through methods like convolutions, restricts the model's ability to adapt its behavior based on the specific content of the input sequence. To address this limitation, Mamba (Gu & Dao, 2023) introduces a selective scan mechanism. While the continuous state evolution matrix $\boldsymbol{A}$ is typically kept fixed and often structured (e.g., diagonal) for stability and computational reasons, other key parameters that govern the discretized SSM dynamics—$\boldsymbol{B}, \boldsymbol{C}$, and the discretization step size $\Delta$—are rendered input-dependent. Specifically, these parameters are dynamically computed from the current input $x_k$, typically via lightweight linear projections or small neural networks. Such an input-dependent formulation, which forms the cornerstone of the full Mamba architecture (Gu & Dao, 2023), empowers the model to selectively emphasize or ignore parts of the input history by dynamically modulating its effective discrete parameters. This allows for the robust modeling of temporal dependencies by modulating state transitions, effectively filtering irrelevant information, and adapting state transitions based on contextual cues within the input.

Building upon these principles, Mamba2 (Dao & Gu, 2024) represents a subsequent refinement, developed under the Structured State Space Duality (SSD) framework, which elucidates deeper connections between SSMs and attention mechanisms. The core of Mamba2 is its SSD layer, where the state transition dynamics $\boldsymbol{A}_k$ are simplified (often to a scalar representation per step), and the layer typically utilizes larger head dimensions (e.g., 64 or 128), aligning more closely with Transformer conventions. These design choices enable the SSD layer to achieve significant speedups, reportedly 2-8x faster than the original Mamba's selective scan, by leveraging hardware matrix multiplication units more effectively. Furthermore, the Mamba-block architecture incorporates changes aimed at improved scalability and system efficiency, such as parallelizing the initial projections that generate the SSM parameters $(\boldsymbol{A}_k, \boldsymbol{B}_k, \boldsymbol{C}_k)$ and the primary input $x_k$. While preserving Mamba's foundational principles of input-dependent selectivity and linear-time processing characteristic, Mamba2 aims to deliver these capabilities with superior computational performance and enhanced amenability to large-scale model training paradigms. We adopt Mamba2 as the forecasting backbone unless otherwise specified, referring to it as Mamba for brevity.

### B.3 MAXIMUM MEAN DISCREPANCY REGULARIZATION

To faithfully reproduce key statistical invariants and attractor geometry, we incorporate regularization terms designed to preserve the distribution of long-term system statistics. According to existing works (Schiff et al., 2024), such a regularization term should satisfy the following properties: (i) it should respect the underlying geometry of the space $\mathcal{A}$ and allow for comparison between measures even when their supports do not overlap; (ii) it must admit an unbiased estimator that can be computed from samples; (iii) it should have low computational complexity relative to both the system's dimensionality and the number of samples; (iv) it needs to guarantee convergence properties on the space of measures defined over $\mathcal{A}$, meaning that if $D(p_1, p) \to 0$, then $p_1$ converges to $p$; and (v) it should achieve parametric rates of estimation, so that the sampling error $|D - \hat{D}|$ does not depend on the dimension of the system.

Integral Probability Metrics (IPMs) (Müller, 1997) provides a general solution for distribution comparison. For any two distributions $p_1$ and $p_2$, IPMs are defined as follows:

$$
\text{IPM}(p_1, p_2) = \sup_{\kappa \in \mathcal{K}} |\mathbb{E}_{\boldsymbol{u} \sim p_1}[\kappa(\boldsymbol{u})] - \mathbb{E}_{\boldsymbol{u}' \sim p_2}[\kappa(\boldsymbol{u}')]|,
\tag{21}
$$

where $\mathcal{K}$ denotes the specific function class. If the function space $\mathcal{K}$ is rich enough, $\mathrm{IPM}(p_1, p_2) \to 0$ indicates $p_1 \to p_2$. Maximum mean discrepancy (MMD), a member of this class, has gained significant attention due to its favorable theoretical and computational properties and ability to satisfy the requirements outlined above. MMD measures the distance between two probability distributions by comparing their embeddings. Specifically, given two distributions $p_1$ and $p_2$ in a reproducing kernel Hilbert space (RKHS). Specifically, given two distributions, the MMD is defined as:

$$\mathrm{MMD}(p_1, p_2) \coloneqq \sup\nolimits_{||f||_{\mathcal{H} \leq 1}} |\mathbb{E}_{\boldsymbol{u} \sim p_1}[f(\boldsymbol{u})] - \mathbb{E}_{\boldsymbol{u'} \sim p_2}[f(\boldsymbol{u'})]|, \tag{22}$$

where $\mathcal{H}$ is an RKHS, and $f(\cdot)$ ranges over functions in $\mathcal{H}$ with norm at most one. This formulation allows for an unbiased estimator that can be efficiently computed from samples drawn from $p_1$ and $p_2$. Moreover, the MMD respects the geometry of the underlying space and can effectively compare distributions even when their supports do not overlap. Using the reproducing property of $\mathcal{H}$ and the Riesz representation theorem, MMD can be represented as follows:

$$\mathrm{MMD}^2(p_1, p_2) = \mathbb{E}_{\boldsymbol{u}, \boldsymbol{u'} \sim p_1}[\kappa(\boldsymbol{u}, \boldsymbol{u'})] + \mathbb{E}_{\boldsymbol{v}, \boldsymbol{v'} \sim p_2}[\kappa(\boldsymbol{v}, \boldsymbol{v'})] - 2\mathbb{E}_{\boldsymbol{u} \sim p_1, \boldsymbol{v} \sim p_2}[\kappa(\boldsymbol{u}, \boldsymbol{v})], \tag{23}$$

where $\kappa : \mathcal{A} \times \mathcal{A} \to \mathbb{R}$ is a kernel function. For the implementation of kernel function $\kappa$, we employ a mixture of rational quadratic kernels following the existing works (Seeger, 2004; Li et al., 2015; Schiff et al., 2024), formulated as:

$$\kappa_{\boldsymbol{\sigma}}(\boldsymbol{u}, \boldsymbol{u'}) = \sum_{\sigma_q \in \boldsymbol{\sigma}} \kappa_{\sigma_q}(\boldsymbol{u}, \boldsymbol{u'}) = \sum_{\sigma_q \in \boldsymbol{\sigma}} \frac{\sigma_q^2}{\sigma_q^2 + ||\boldsymbol{u} - \boldsymbol{v}||_2^2}, \tag{24}$$

where $\boldsymbol{\sigma} = \{0.2, 0.5, 0.9, 1.3\}$, which is also consistent with existing works.

## C  EXPERIMENTAL DETAILS

### C.1  DETAILS OF DATASETS

**Lorenz63 system.** The Lorenz63 system is a 3-dimensional system defined by the following ordinary differential equations:

$$\begin{aligned} \dot{x} &= \sigma(y - x) \\ \dot{y} &= \rho x - y - xz \\ \dot{z} &= xy - \beta z \end{aligned} \tag{25}$$

The parameter values are $\sigma = 10, \rho = 28, \beta = 8/3$, which is known to show the chaotic behavior with a strange attractor (Tucker, 2002; Luzzatto et al., 2005).

**Rossler system.** The Rossler system is a 3-dimensional system defined by the following ordinary differential equations:

$$\begin{aligned} \dot{x} &= -(y + z) \\ \dot{y} &= x + \alpha y \\ \dot{z} &= \beta + z(x - \gamma) \end{aligned} \tag{26}$$

The parameters are taken to be typical values, i.e., $\alpha = 0.2, \beta = 0.2, \gamma = 5.7$, which can produce a strange attractor.

**Lorenz96 system.** The Lorenz96 system is defined by the following ordinary differential equations:

$$\dot{x}_k = (x_{k+1} - x_{k-2})x_{k-1} - x_k + F, \quad k = 1, 2, \ldots, N, \tag{27}$$

where $N$ denotes the dimension of the system. The system can be defined on arbitrarily high dimensions. Without loss of generality, we take $N = 5$ and $F = 20$, such that the system can be located within the chaotic regime.

We generate data for the above three datasets with the fourth-order Runge-Kutta numerical integrator with $\Delta t = 0.001$ based on `dyst` (Gilpin, 2021) library. In order to unify the time granularity of different systems, after generating the trajectory, we downsample the resulting trajectory to a uniform granularity of about 30 time points per Lyapunov time.

For each system, we generate a sequence with 30000 time points (about 1000 Lyapunov times), and use a sliding window with 30 time points (about 1 Lyapunov time) to generate the training

samples. During the teacher-forcing training phase, we shuffle the generated training samples and ensure that each sample contains only 30 time steps, corresponding to approximately one Lyapunov time. As a result, these samples capture temporal dynamics over a relatively short horizon, making them challenging for training models aimed at long-term forecasting. During the student forcing training phase, we use a sliding window with 60 time points (about 2 Lyapunov times). The model employs the first 30 time points as the historical context and predicts the last 30 time points in an autoregressive manner. For baseline models, we use the dataset of the student forcing training phase for training. The validation trajectory contains 150 time points, and we use the same split method to generate the validation data for the two training stages. We generate the test dataset for each system with 100 different initial conditions to evaluate whether the model can capture the intrinsic dynamics of the underlying system. Each test sample contains 30 time steps (1 Lyapunov time) as historical observations, and 300 time steps (10 Lyapunov times) as forecasting objectives.

**Electrocardiogram (ECG) dataset.** We adopt the Electrocardiogram (ECG) dataset from existing works (Mikhaeil et al., 2022). ECG dataset records a scalar physiological (heart muscle potential) time series, and it is embedded into a 5-dimensional space using the PECUZAL algorithm (Krämer et al., 2021). The data contains two trajectories, and each trajectory contains 100,000 time points ($\approx 143s$). The maximum Lyapunov exponent is estimated as $\lambda_{\max} = (2.19 \pm 0.05)\frac{1}{\text{s}}$, which is consistent with the literature (Govindan et al., 1998; Mikhaeil et al., 2022). We resample two trajectories with an interval of 10 time steps, and the granularity of the resulting trajectory is about 30 time points per Lyapunov time. We use one of the trajectories to construct the training and validation datasets, and the construction process is the same as the above three simulation systems. We split another trajectory with a sliding window of 500 time steps. For each sliding window, the last 300 time steps (about 10 Lyapunov times) are treated as forecasting objectives. The model takes the preceding 30 time steps (about 1 Lyapunov time) as historical observations. The remaining 170 time steps are treated as intervals to ensure the difference between test samples. After processing, we derive 20 test samples.

**Electroencephalogram (EEG) dataset.** We adopt the Electroencephalogram (EEG) dataset from existing works (Mikhaeil et al., 2022; Hess et al., 2023). EEG records are from a human subject under task-free, quiescent conditions with eyes open. The dataset contains two trajectories, and each trajectory contains 9,640 time points. The maximum Lyapunov exponent is estimated as $\lambda_{\max} \approx 0.017$ ($\approx 60$ time steps per Lyapunov time), which is consistent with the literature (Mikhaeil et al., 2022; Hess et al., 2023). Each trajectory represents brain activity over a 60-second interval, recorded using 64 electrodes distributed across the scalp, resulting in a 64-dimensional time series. Due to the limited number of time steps, no further resampling is performed. One of the trajectories is used to construct the training and validation datasets, following the same procedure as for the three simulated systems described above, except that one Lyapunov time corresponds to 60 time steps in this case. We split another trajectory with a sliding window of 700 time steps. For each sliding window, the last 600 time steps (about 10 Lyapunov times) are treated as forecasting objectives. The model takes the preceding 60 time steps (about 1 Lyapunov time) as historical observations. The remaining 30 time steps are treated as intervals to ensure the difference between test samples. After processing, we derive 13 test samples.

## C.2  DETAILS OF EVALUATION METRICS

- **1-step Error.** We employ mean absolute error (MAE) to measure the forecasting accuracy between $\boldsymbol{x}_{t+1}$ and $\hat{\boldsymbol{x}}_{t+1}$ at the immediate following step.

- **sMAPE.** With regard to short-term predictive accuracy, we utilize the Symmetric Mean Absolute Percentage Error (SMAPE) to evaluate point-wise forecasting accuracy, consistent with existing works (Zhang & Gilpin, 2024). It provides a robust measure of relative error that is less sensitive to outliers compared to traditional metrics like Mean Absolute Error, defined as:

$$\text{sMAPE} \equiv 2\frac{100}{T} \sum_{t=1}^{T} \frac{|\boldsymbol{x}_t - \hat{\boldsymbol{x}}_t|}{|\boldsymbol{x}_t| + |\hat{\boldsymbol{x}}_t|}, \tag{28}$$

  where $\boldsymbol{x}_t$ and $\hat{\boldsymbol{x}}_t$ denote the true and predicted values of system states at time step $t$, and $T$ corresponds to the total evaluation steps.

- **VPT.** We also incorporate another metric, valid prediction time (VPT), the first time step where sMAPE exceeds a pre-defined threshold $\epsilon$, formulated as:

$$\text{VPT} \equiv \text{argmax}_{t_f} \{t_f | \text{sMAPE}(\boldsymbol{x}_t, \hat{\boldsymbol{x}}_t) < \epsilon, \forall t < t_f\}. \tag{29}$$

Table 3: Hyperparameter settings of PhyxMamba on simulated and real-world datasets. TF and SF represent teacher forcing and student forcing, respectively. Hidden size, expand, and size per head are internal hyperparameters of Mamba blocks.

| Parameter / System | $D$ | $d$ | $m$ | $\tau$ | $M$ | $L$ | $\lambda_p$ | $\lambda_c$ | $\lambda_\tau$ | TF-$lr$ | SF-$lr$ | Hidden size | Expand | Size per head |
|---|---|---|---|---|---|---|---|---|---|---|---|---|---|---|
| Lorenz63 (3D) | 10 | 256 | 3 | 7 | 3 | 4 | 0.1 | 1000 | 1 | 0.001 | 0.0001 | 1024 | 2 | 64 |
| Rossler (3D) | 5 | 256 | 3 | 7 | 3 | 1 | 0.1 | 1000 | 1 | 0.001 | 0.0001 | 1024 | 2 | 64 |
| Lorenz96 (5D) | 5 | 256 | 4 | 3 | 3 | 3 | 0.1 | 1000 | 1 | 0.001 | 0.0001 | 1024 | 2 | 64 |
| ECG (5D) | 2 | 256 | 4 | 6 | 3 | 4 | 0.1 | 1000 | 1 | 0.001 | 0.0001 | 1024 | 2 | 64 |
| EEG (64D) | 12 | 256 | 4 | 9 | 3 | 4 | 0.1 | 1000 | 1 | 0.001 | 0.0001 | 1024 | 2 | 64 |

Consistent the standard practice established in existing works on chaotic system forecasting (Gilpin, 2023; Zhang & Gilpin, 2024; Vlachas et al., 2020), without loss of generality, we set $\epsilon = 30$ in our experiments. While the absolute VPT values are dependent on this threshold, the relative performance and ranking among the different models are robust to this choice. A different value for $\epsilon$ would scale the VPTs for all methods.

- **Correlation Dimension Error** ($D_{\text{frac}}$). Besides metrics focusing on point-wise prediction accuracy, we introduce this metric to evaluate the model's capability to replicate the geometric complexity of the system's strange attractor. The correlation dimension is a measure of the fractal dimension, a unique and invariant property of the strange attractor, quantifying the spatial correlation of points on it. It provides insight into the degrees of freedom effectively active in the system's dynamics. We calculate the correlation dimension for both the true attractor, derived from the ground-truth data, and the attractor generated from the model's long-term forecasted trajectory. The Correlation Dimension Error is then defined as the root mean square error between them. A smaller error signifies that the forecasted dynamics faithfully reproduce the intrinsic geometric structure and complexity of the true strange attractor, which is a critical aspect of long-term chaotic system forecasting.

- **KL Divergence between Attractors** ($D_{\text{stsp}}$). To evaluate the fidelity between the true attractor and the one reconstructed by the model, we employ the Kullback-Leibler (KL) divergence, following previous works (Hess et al., 2023; Göring et al., 2024; Zhang & Gilpin, 2024). The long-term dynamics of a chaotic system are characterized by a probability distribution over its phase space, which quantifies the likelihood of the system residing in any particular state. Operationally, we approximate these invariant distributions for both the true and reconstructed attractors by constructing Gaussian mixture models derived from points sampled along their respective trajectories. The KL divergence between these two mixture distributions is then estimated using a sampling-based approach. Consequently, a lower KL divergence value indicates that the reconstructed attractor more faithfully captures the statistical properties and phase-space density distribution of the true system's attractor.

## C.3 DETAILS OF HYPERPARAMETER SETTINGS

We demonstrate the hyperparameter settings of our model on simulated and real-world datasets in Table 3. All experiments are completed on a single NVIDIA RTX 4090 GPU.

## C.4 DETAILS OF ABLATION STUDY IMPLEMENTATIONS

In this section, we remove designs of our models to demonstrate their effectiveness on the improvement of forecasting performance.

**Encoder-oriented Mamba Variant.** We design an encoder-oriented Mamba architecture to evaluate the effectiveness of our generative training strategy. Specifically, we use Mamba blocks as the encoder, where historical observations are input to produce an encoded representation $\boldsymbol{Z} \in \mathbb{R}^{T \times d}$. This representation is then flattened to $\mathbb{R}^{T*d}$ and passed to a decoder composed of linear layers. During training, the model forecasts the next $T$ steps during the training stage. time steps in a single forward pass. In the inference stage, long-term forecasting is performed in an autoregressive manner, where each predicted sequence of $T$ steps is recursively used as the input context for subsequent predictions.

**w/o PIR.** We bypass the physics-guided representation framework—specifically, the time-delay embedding module—and proceed directly to the patching and embedding process.

**w/o RS.** We replace the hierarchical Mamba block—specifically, the residual stacked variant—with a layer-by-layer stacked Mamba block to assess the effectiveness of decomposed forecasting.

**w/o MPP.** We remove the multi-patch prediction objective and only retain the fundamental next-patch prediction strategy for training the forecasting backbone.

**w/o SF.** We remove the student forcing training stage and only retain the teacher forcing training stage.

**w/o MMD.** We remove the maximum mean discrepancy (MMD) regularization objective in the student forcing stage.

# D    DISCUSSIONS

## D.1    CODE OF ETHICS

The use of open-sourced and publicly available models and datasets strictly adheres to their respective licenses. All of them are respectfully cited in Section 1 and Section 4 of the main text.

## D.2    BROADER IMPACT

This research on long-term chaotic system forecasting from short-term data has significant potential for broad societal benefits. The proposed PhyxMamba framework, by reliably predicting chaotic systems under observation-scarce conditions, could have wide-ranging implications across various critical domains. For instance, in climate science, improved forecasting could lead to better preparedness for extreme weather events. In neuroscience, it could aid in understanding complex brain dynamics and potentially contribute to diagnosing and treating neurological disorders. Furthermore, in epidemiology, more accurate long-term predictions of disease spread could inform public health policies and resource allocation. The ability to make informed decisions based on better understanding and forecasting of such pervasive systems is crucial. Ultimately, this work opens new avenues for modeling complex dynamical systems, which could drive advancements in numerous scientific and engineering disciplines.

## D.3    USAGE OF LARGE LANGUAGE MODEL DECLARATION

The authors hereby declare the use of the Large Language Model (LLM) during the preparation of this paper. The role of the LLM is exclusively confined to language polishing and refinement of the manuscript's expression. All foundational and critical aspects of the research, including the formulation of the core ideas, the design of the proposed scheme, the planning of experiments, and the acquisition and analysis of all experimental data, are conducted without the assistance of any AI-based tools and are the sole contribution of the authors.

# E    EXTENDED RESULTS

## E.1    DETAILED RESULTS OF OVERALL PERFORMANCE

Here we provide a full table of model performances on five simulated and real-world datasets, including the results of 95% confidence interval, in Table 4. We also show the visualization results on the Lorenz96 system (first 3-dimension), the ECG dataset, and the EEG dataset in Figure 6, Figure 7, and Figure 8, respectively. Furthermore, we provide a dimension-wise comparison between our predictions and the ground truth for all systems in Figure 9. We reduce the dimensionality of the high-dimensional system *i.e.,* Lorenz96, ECG, and EEG, to three dimensions using Principal Component Analysis (PCA) to facilitate this visualization.

Table 4: Overall performance on both simulated and real-world datasets. Reported values represent the mean ± 95% confidence interval (CI), where CIs are computed based on multiple runs from different randomly sampled initial conditions to capture variability in the model's predictions. The best performance of each metric is marked in **bold**, and the second-best performance is underlined.

| System | Metric | Ours | NBEATS | NHITS | CrossFormer | PatchTST | TimesNet | TiDE | iTransformer | Koopa | PLRNN | nVAR | PRC | HsGRC | DLinear | NSFormer | FEDFormer | AutoFormer | Timet(84M) | Chronos(205M) | Chronos(48M) | Chronos(21M) | Chronos(9M) |
|---|---|---|---|---|---|---|---|---|---|---|---|---|---|---|---|---|---|---|---|---|---|---|---|
| Lorenz63 (3D) | VPT (↑) | 5.06 ± 0.38 | 3.66 ± 0.31 | 2.86 ± 0.21 | 1.95 ± 0.27 | 0.46 ± 0.17 | 1.01 ± 0.05 | 0.32 ± 0.14 | 0.30 ± 0.12 | 0.33 ± 0.15 | 0.71 ± 0.12 | 0.20 ± 0.03 | 1.59 ± 0.26 | 1.12 ± 0.012 | 0.19 ± 0.05 | 0.90 ± 0.19 | 0.37 ± 0.08 | 0.23 ± 0.08 | 0.06 ± 0.02 | 0.20 ± 0.058 | 0.14 ± 0.03 | 0.11 ± 0.03 | 0.09 ± 0.02 |
|  | 1-step Error (↓) | 0.083 ± 0.011 | 0.056 ± 0.001 | 0.252 ± 0.03 | 0.605 ± 0.078 | 1.299 ± 0.18 | 0.389 ± 0.040 | 0.711 ± 0.101 | 1.271 ± 0.335 | 2.600 ± 0.229 | 0.380 ± 0.039 | 0.402 ± 0.089 | 0.0004 ± 0.0000 | 0.0002 ± 0.0000 | 2.318 ± 0.251 | 1.796 ± 0.205 | 1.524 ± 0.221 | 1.998 ± 0.210 | 0.460 ± 0.178 | 1.020 ± 0.142 | 1.336 ± 0.167 | 1.573 ± 0.185 | 1.984 ± 0.210 |
|  | sMAPE@1 (↓) | 5.36 ± 1.56 | 5.14 ± 0.81 | 5.34 ± 0.81 | 13.62 ± 1.67 | 64.39 ± 0.40 | 35.84 ± 4.94 | 84.88 ± 0.99 | 67.84 ± 6.55 | 60.68 ± 5.75 | 59.26 ± 6.79 | 94.32 ± 4.32 | 18.34 ± 1.91 | 93.32 ± 0.57 | 89.64 ± 7.52 | 39.49 ± 3.91 | 54.15 ± 5.41 | 52.33 ± 5.20 | 121.42 ± 4.86 | 89.19 ± 5.58 | 87.83 ± 5.00 | 91.56 ± 4.98 | 92.89 ± 4.76 |
|  | sMAPE@64 (↓) | 22.05 ± 3.13 | 37.08 ± 3.10 | 52.36 ± 3.78 | 64.40 ± 4.48 | 94.39 ± 4.61 | 103.25 ± 2.72 | 102.83 ± 3.00 | 107.96 ± 3.12 | 100.54 ± 3.14 | 104.43 ± 2.69 | 110.23 ± 1.57 | 135.42 ± 5.78 | 102.96 ± 0.32 | 106.63 ± 4.33 | 103.81 ± 4.21 | 95.67 ± 3.11 | 104.89 ± 2.04 | 167.67 ± 2.23 | 107.66 ± 2.48 | 109.97 ± 2.36 | 105.45 ± 3.22 | 104.45 ± 3.14 |
|  | sMAPE@10 (↓) | 67.29 ± 3.49 | 78.46 ± 2.78 | 84.18 ± 2.43 | 92.96 ± 2.84 | 104.24 ± 3.17 | 0.309 ± 0.029 | 0.696 ± 0.049 | 0.638 ± 0.016 | 0.333 ± 0.049 | 0.270 ± 0.014 | 0.792 ± 0.009 | 0.578 ± 0.049 | 0.099 ± 0.011 | 0.565 ± 0.008 | 0.430 ± 0.036 | 0.115 ± 0.014 | 0.247 ± 0.019 | 0.311 ± 0.012 | 0.264 ± 0.020 | 0.188 ± 0.022 | 0.239 ± 0.024 | 0.217 ± 0.027 |
|  | $D_{stsp}$ (↓) | 0.060 ± 0.008 | 0.075 ± 0.008 | 0.067 ± 0.012 | 0.09 ± 0.012 | 0.227 ± 0.034 | 12.34 ± 1.371 | 12.34 ± 2.55 | 62.44 ± 3.378 | 83.768 ± 2.55 | 12.970 ± 4.578 | 61.301 ± 2.846 | 29.889 ± 2.35 | 0.099 ± 0.011 | 52.023 ± 1.62 | 31.378 ± 3.793 | 16.701 ± 4.533 | 15.066 ± 1.952 | 43.705 ± 1.778 | 116.88 ± 1.468 | 41.786 ± 3.720 | 41.282 ± 3.755 | 40.094 ± 4.066 |
|  | $D_{H}$ (↓) | 1.133 ± 0.340 | 1.566 ± 0.354 | 1.648 ± 0.172 | 1.591 ± 0.132 | 1.644 ± 0.382 |  |  |  |  |  |  |  |  |  |  |  |  |  |  |  |  |  |
| Rossler (3D) | VPT (↑) | 9.83 ± 0.20 | 6.48 ± 0.41 | 0.44 ± 0.12 | 0.95 ± 0.14 | 0.04 ± 0.01 | 0.26 ± 0.08 | 0.04 ± 0.00 | 0.07 ± 0.02 | 0.05 ± 0.00 | 0.44 ± 0.16 | 0.01 ± 0.01 | 1.13 ± 0.07 | 0.83 ± 0.18 | 0.00 ± 0.00 | 0.06 ± 0.02 | 0.01 ± 0.00 | 0.01 ± 0.00 | 0.00 ± 0.00 | 0.02 ± 0.00 | 0.03 ± 0.01 | 0.00 ± 0.00 | 0.01 ± 0.00 |
|  | 1-step Error (↓) | 0.007 ± 0.001 | 14.88 ± 0.79 | 39.90 ± 1.56 | 0.092 ± 0.041 | 0.422 ± 0.050 | 0.102 ± 0.035 | 0.382 ± 0.012 | 0.659 ± 0.136 | 0.677 ± 0.149 | 0.050 ± 0.016 | 0.288 ± 0.100 | 0.0009 ± 0.0001 | 0.0036 ± 0.0005 | 0.800 ± 0.072 | 0.483 ± 0.03 | 0.428 ± 0.037 | 0.586 ± 0.051 | 1.763 ± 0.12 | 0.423 ± 0.057 | 0.435 ± 0.068 | 0.558 ± 0.079 | 0.712 ± 0.125 |
|  | sMAPE@1 (↓) | 2.84 ± 0.18 | 22.68 ± 0.73 | 64.07 ± 2.80 | 27.11 ± 1.25 | 89.16 ± 3.45 | 42.48 ± 2.87 | 98.48 ± 3.42 | 85.85 ± 5.04 | 67.78 ± 3.65 | 51.36 ± 4.48 | 94.95 ± 3.20 | 28.38 ± 2.75 | 31.03 ± 1.88 | 102.02 ± 3.17 | 52.33 ± 3.45 | 67.14 ± 2.05 | 65.42 ± 1.89 | 179.74 ± 2.63 | 80.91 ± 3.49 | 95.68 ± 2.67 | 86.47 ± 3.28 | 103.69 ± 3.48 |
|  | sMAPE@64 (↓) | 3.55 ± 0.17 | 83.54 ± 2.43 | 83.54 ± 2.43 | 38.32 ± 1.12 | 111.19 ± 2.32 | 107.31 ± 2.55 | 116.59 ± 2.08 | 120.87 ± 2.55 | 99.61 ± 2.78 | 84.09 ± 5.49 | 115.42 ± 2.15 | 120.49 ± 2.85 | 110.01 ± 6.25 | 120.49 ± 2.85 | 120.40 ± 2.95 | 94.19 ± 2.55 | 100.21 ± 2.45 | 186.70 ± 1.02 | 108.03 ± 2.84 | 123.16 ± 1.96 | 86.83 ± 2.37 | 129.54 ± 2.50 |
|  | sMAPE@10 (↓) | 12.64 ± 1.94 | 14.12 ± 0.56 | 83.54 ± 2.43 | 62.43 ± 1.85 | 129.67 ± 2.20 | 138.19 ± 2.21 | 126.01 ± 2.79 | 128.68 ± 2.38 | 122.36 ± 2.40 | 107.07 ± 6.28 | 140.00 ± 1.11 | 139.02 ± 1.77 | 115.82 ± 3.03 | 140.49 ± 1.11 | 141.79 ± 2.68 | 106.07 ± 1.70 | 122.18 ± 2.49 | 187.83 ± 0.94 | 127.75 ± 2.50 | 136.11 ± 1.42 | 133.56 ± 3.18 | 130.67 ± 1.78 |
|  | $D_{stsp}$ (↓) | 0.049 ± 0.008 | 0.069 ± 0.009 | 0.088 ± 0.010 | 0.074 ± 0.011 | 0.158 ± 0.012 | 0.530 ± 0.028 | 0.455 ± 0.014 | 0.383 ± 0.030 | 0.497 ± 0.060 | 0.400 ± 0.015 | 0.750 ± 0.013 | 0.770 ± 0.022 | 0.611 ± 0.027 | 0.184 ± 0.012 | 0.389 ± 0.041 | 0.106 ± 0.014 | 0.207 ± 0.019 | 0.535 ± 0.017 | 0.765 ± 0.020 | 0.114 ± 0.015 | 0.217 ± 0.015 | 0.095 ± 0.012 |
|  | $D_{H}$ (↓) | 0.034 ± 0.009 | 0.117 ± 0.023 | 0.538 ± 0.021 | 0.187 ± 0.033 | 8.423 ± 0.031 | 4.995 ± 0.64 | 15.034 ± 0.976 | 10.807 ± 0.929 | 5.721 ± 0.316 | 3.950 ± 0.481 | 7.391 ± 0.632 | 4.549 ± 0.579 | 4.068 ± 0.214 | 10.752 ± 0.438 | 4.642 ± 0.293 | 0.833 ± 0.076 | 0.989 ± 0.087 | 121.65 ± 0.566 | 4.397 ± 0.445 | 3.369 ± 0.240 | 4.485 ± 0.259 | 4.813 ± 0.258 |
| Lorenz96 (5D) | VPT (↑) | 1.66 ± 0.19 | 0.92 ± 0.07 | 0.61 ± 0.05 | 0.28 ± 0.05 | 0.05 ± 0.01 | 0.06 ± 0.03 | 0.06 ± 0.01 | 0.02 ± 0.00 | 0.00 ± 0.00 | 0.23 ± 0.03 | 0.17 ± 0.05 | 0.99 ± 0.12 | 0.68 ± 0.16 | 0.01 ± 0.00 | 0.01 ± 0.01 | 0.01 ± 0.01 | 0.01 ± 0.01 | 0.00 ± 0.00 | 0.04 ± 0.02 | 0.04 ± 0.01 | 0.03 ± 0.00 | 0.02 ± 0.00 |
|  | 1-step Error (↓) | 0.155 ± 0.015 | 0.100 ± 0.010 | 0.071 ± 0.009 | 0.583 ± 0.048 | 0.948 ± 0.063 | 1.020 ± 0.093 | 0.480 ± 0.075 | 1.596 ± 0.217 | 2.880 ± 0.222 | 0.230 ± 0.024 | 0.519 ± 0.069 | 0.056 ± 0.006 | 0.097 ± 0.004 | 1.765 ± 0.157 | 2.662 ± 0.243 | 1.923 ± 0.127 | 2.554 ± 0.196 | 0.710 ± 0.073 | 0.994 ± 0.078 | 1.221 ± 0.111 | 1.352 ± 0.114 | 1.560 ± 0.125 |
|  | sMAPE@1 (↓) | 20.28 ± 2.69 | 32.51 ± 4.32 | 51.37 ± 3.98 | 56.64 ± 2.55 | 110.65 ± 2.67 | 99.93 ± 2.52 | 123.41 ± 2.71 | 126.08 ± 2.80 | 114.74 ± 2.51 | 89.90 ± 4.18 | 86.04 ± 1.98 | 36.42 ± 2.36 | 47.75 ± 2.33 | 146.56 ± 1.57 | 104.12 ± 2.62 | 101.29 ± 2.51 | 98.70 ± 2.62 | 142.80 ± 1.85 | 113.16 ± 2.24 | 117.89 ± 2.19 | 160.09 ± 2.40 | 121.85 ± 2.28 |
|  | sMAPE@64 (↓) | 81.18 ± 4.45 | 100.46 ± 3.30 | 105.73 ± 2.88 | 11.21 ± 1.70 | 130.39 ± 1.97 | 122.15 ± 1.52 | 128.61 ± 2.10 | 128.10 ± 2.23 | 127.03 ± 2.29 | 124.11 ± 2.22 | 123.77 ± 2.88 | 102.44 ± 3.78 | 106.73 ± 4.33 | 160.20 ± 0.66 | 124.28 ± 1.60 | 123.93 ± 1.91 | 126.14 ± 1.60 | 142.80 ± 1.25 | 118.54 ± 1.11 | 121.89 ± 1.25 | 122.71 ± 1.56 | 128.67 ± 1.38 |
|  | sMAPE@10 (↓) | 113.56 ± 2.37 | 117.50 ± 1.25 | 125.51 ± 1.74 | 124.31 ± 0.96 | 126.2 ± 0.031 | 127.19 ± 1.64 | 130.62 ± 2.20 | 129.23 ± 2.11 | 130.06 ± 2.28 | 132.05 ± 1.36 | 128.39 ± 1.64 | 116.02 ± 1.91 | 130.34 ± 2.21 | 163.48 ± 0.35 | 128.39 ± 1.64 | 127.53 ± 1.76 | 133.79 ± 1.19 | 133.79 ± 1.19 | 121.73 ± 1.11 | 121.33 ± 1.11 | 127.76 ± 1.75 | 130.39 ± 1.23 |
|  | $D_{stsp}$ (↓) | 0.154 ± 0.030 | 0.172 ± 0.023 | 0.195 ± 0.024 | 0.196 ± 0.034 | 1.262 ± 0.033 | 1.172 ± 0.028 | 1.273 ± 0.027 | 1.252 ± 0.026 | 0.806 ± 0.036 | 0.270 ± 0.019 | 0.283 ± 0.027 | 0.179 ± 0.036 | 0.154 ± 0.010 | 1.258 ± 0.023 | 1.002 ± 0.035 | 0.219 ± 0.023 | 0.269 ± 0.031 | 0.598 ± 0.031 | 0.630 ± 0.039 | 0.740 ± 0.030 | 0.797 ± 0.042 | 0.659 ± 0.039 |
|  | $D_{H}$ (↓) | 18.446 ± 0.799 | 11.083 ± 1.000 | 18.132 ± 0.727 | 36.191 ± 1.313 | 102.444 ± 5.324 | 102.444 ± 5.324 | 74.493 ± 4.897 | 170.792 ± 5.210 | 120.611 ± 4.192 | 243.020 ± 1.655 | 237.557 ± 0.727 | 16.834 ± 0.839 | 17.287 ± 0.669 | 226.606 ± 2.925 | 111.383 ± 3.127 | 61.235 ± 2.635 | 57.292 ± 2.735 | 137.37 ± 2.442 | 921.49 ± 2.894 | 97.201 ± 2.659 | 92.927 ± 2.954 | 81.569 ± 2.815 |
| ECG (5D) | VPT (↑) | 0.93 ± 0.45 | 0.61 ± 0.16 | 0.59 ± 0.23 | 0.33 ± 0.20 | 0.29 ± 0.19 | 0.32 ± 0.32 | 0.01 ± 0.00 | 0.51 ± 0.29 | 0.21 ± 0.16 | 0.20 ± 0.00 | 0.04 ± 0.025 | 0.398 ± 0.25 | 0.19 ± 0.09 | 0.437 ± 0.185 | 0.183 ± 0.076 | 0.298 ± 0.099 | 0.215 ± 0.065 | 0.004 ± 0.002 | 0.361 ± 0.218 | 0.411 ± 0.217 | 0.432 ± 0.215 | 0.426 ± 0.223 |
|  | 1-step Error (↓) | 0.017 ± 0.006 | 0.024 ± 0.001 | 0.024 ± 0.001 | 0.070 ± 0.021 | 0.072 ± 0.037 | 0.083 ± 0.030 | 0.316 ± 0.139 | 0.063 ± 0.025 | 0.087 ± 0.032 | 0.032 ± 0.016 | 0.092 ± 0.051 | 136.60 ± 12.07 | 44.10 ± 6.51 | 146.07 ± 9.44 | 37.22 ± 7.68 | 96.52 ± 19.68 | 76.61 ± 16.34 | 159.99 ± 9.01 | 137.98 ± 10.23 | 142.13 ± 9.09 | 148.20 ± 8.52 | 133.18 ± 10.45 |
|  | sMAPE@1 (↓) | 37.14 ± 15.75 | 87.67 ± 16.03 | 84.19 ± 11.07 | 58.70 ± 17.62 | 64.83 ± 18.91 | 60.84 ± 16.79 | 136.53 ± 6.10 | 55.59 ± 17.30 | 60.63 ± 17.08 | 146.00 ± 13.33 | 147.87 ± 5.04 | 154.30 ± 3.83 | 159.76 ± 6.07 | 159.73 ± 7.12 | 135.25 ± 5.94 | 95.76 ± 4.37 | 139.70 ± 4.40 | 138.08 ± 9.63 | 149.35 ± 5.74 | 147.38 ± 6.55 | 145.05 ± 6.73 | 140.13 ± 5.81 |
|  | sMAPE@64 (↓) | 73.64 ± 14.33 | 198.39 ± 5.43 | 84.19 ± 11.07 | 101.15 ± 11.29 | 124.47 ± 13.37 | 152.53 ± 9.03 | 139.87 ± 9.04 | 117.09 ± 13.23 | 122.57 ± 10.23 | 145.26 ± 12.80 | 183.33 ± 3.59 | 151.54 ± 5.63 | 95.76 ± 6.80 | 154.30 ± 3.36 | 132.52 ± 5.94 | 154.30 ± 3.83 | 124.75 ± 5.00 | 169.72 ± 5.08 | 158.08 ± 9.63 | 147.82 ± 6.20 | 142.27 ± 6.95 | 140.14 ± 5.56 |
|  | sMAPE@10 (↓) | 99.68 ± 9.68 | 108.26 ± 9.20 | 84.39 ± 12.07 | 124.75 ± 2.92 | 156.24 ± 6.62 | 132.05 ± 7.80 | 128.69 ± 9.82 | 47.44 ± 6.71 | 151.04 ± 5.63 | 152.53 ± 4.08 | 45.85 ± 6.67 | 172.40 ± 4.36 | 184.69 ± 1.33 | 144.70 ± 4.36 | 172.40 ± 4.36 | 159.76 ± 6.00 | 144.70 ± 4.36 | 92.07 ± 4.81 | 147.52 ± 6.55 | 147.82 ± 6.20 | 147.38 ± 6.20 | 140.61 ± 5.56 |
|  | $D_{stsp}$ (↓) | 0.036 ± 0.013 | 0.078 ± 0.012 | 0.078 ± 0.012 | 0.443 ± 0.050 | 0.248 ± 0.048 | 0.143 ± 0.059 | 0.219 ± 0.021 | 0.292 ± 0.016 | 0.245 ± 0.070 | 0.529 ± 0.102 | 0.315 ± 0.061 | 0.448 ± 0.027 | 0.184 ± 0.025 | 0.170 ± 0.023 | 0.245 ± 0.070 | 0.738 ± 0.064 | 0.443 ± 0.031 | 0.443 ± 0.060 | 0.443 ± 0.097 | 0.443 ± 0.086 | 0.381 ± 0.086 | 0.362 ± 0.067 |
|  | $D_{H}$ (↓) | 0.004 ± 0.001 | 0.034 ± 0.005 | 0.035 ± 0.005 | 0.156 ± 0.035 | 0.440 ± 0.079 | 0.682 ± 0.104 | 0.834 ± 0.076 | 0.459 ± 0.082 | 0.463 ± 0.068 | 1.992 ± 0.934 | 0.638 ± 0.053 | 3.265 ± 1.720 | 4.015 ± 0.822 | 0.670 ± 0.041 | 1.005 ± 0.212 | 1.215 ± 0.168 | 0.806 ± 0.149 | 0.675 ± 2.442 | 0.480 ± 0.079 | 0.542 ± 0.090 | 0.511 ± 0.072 | 0.507 ± 0.086 |
| EEG (64D) | VPT (↑) | 9.43 ± 1.24 | 0.083 ± 0.1 | 0.253 ± 0.470 | 0.023 ± 0.058 | 0.025 ± 0.039 | 0.01 ± 0.00 | 0.036 ± 0.032 | 0.015 ± 0.031 | 0.001 ± 0.000 | 0.029 ± 0.007 | 0.00 ± 0.00 | 0.035 ± 0.027 | 0.19 ± 0.09 | 0.035 ± 0.009 | 0.008 ± 0.005 | 0.000 ± 0.000 | 0.000 ± 0.000 | 0.10 ± 0.03 | 0.06 ± 0.02 | 0.06 ± 0.012 | 0.05 ± 0.01 | 0.058 ± 0.012 |
|  | 1-step Error (↓) | 0.031 ± 0.003 | 0.036 ± 0.004 | 0.035 ± 0.003 | 0.169 ± 0.061 | 0.068 ± 0.015 | 0.562 ± 0.059 | 0.088 ± 0.019 | 0.201 ± 0.053 | 0.257 ± 0.069 | 0.087 ± 0.0037 | 1.063 ± 0.034 | 0.099 ± 0.010 | 0.031 ± 0.013 | 0.234 ± 0.054 | 0.353 ± 0.055 | 0.314 ± 0.105 | 0.478 ± 0.086 | 0.023 ± 0.006 | 0.042 ± 0.010 | 0.054 ± 0.012 | 0.090 ± 0.010 | 0.058 ± 0.012 |
|  | sMAPE@1 (↓) | 14.21 ± 2.06 | 79.28 ± 18.19 | 72.03 ± 13.32 | 122.58 ± 11.92 | 72.08 ± 11.67 | 126.52 ± 12.60 | 116.12 ± 12.60 | 120.88 ± 4.89 | 125.12 ± 12.60 | 147.84 ± 14.79 | 145.99 ± 14.53 | 80.102 ± 17.74 | 80.102 ± 17.74 | 124.64 ± 14.77 | 191.03 ± 9.54 | 125.65 ± 9.42 | 126.27 ± 11.37 | 125.65 ± 15.61 | 166.88 ± 15.16 | 17.98 ± 15.49 | 18.45 ± 15.49 | 179.78 ± 16.67 |
|  | sMAPE@64 (↓) | 18.12 ± 9.35 | 130.69 ± 8.04 | 98.01 ± 11.33 | 144.06 ± 10.58 | 125.94 ± 9.25 | 148.73 ± 6.67 | 139.01 ± 9.83 | 140.82 ± 11.55 | 144.83 ± 6.65 | 168.68 ± 11.55 | 148.98 ± 10.19 | 194.70 ± 14.55 | 194.70 ± 14.55 | 138.43 ± 11.33 | 142.16 ± 10.34 | 144.61 ± 7.66 | 130.70 ± 4.40 | 134.75 ± 9.18 | 138.08 ± 9.63 | 138.72 ± 9.50 | 38.46 ± 9.70 | 137.56 ± 10.34 |
|  | sMAPE@10 (↓) | 21.87 ± 16.93 | 131.69 ± 5.50 | 136.07 pm 8.04 | 154.63 ± 6.77 | 145.33 ± 6.83 | 145.86 ± 7.20 | 144.80 ± 7.89 | 146.10 ± 8.21 | 143.83 ± 6.65 | nan ± nan | 149.11 ± 11.04 | 144.52 ± 8.10 | 98.97 ± 14.32 | 178.17 ± 2.08 | 146.16 ± 7.76 | 16.91 ± 5.99 | 14.78 ± 4.70 | 162.91 ± 7.42 | 142.16 ± 7.62 | 142.20 ± 7.20 | 142.09 ± 7.15 | 14.137 ± 6.91 |
|  | $D_{stsp}$ (↓) | 0.180 ± 0.106 | 0.620 ± 0.202 | 0.501 ± 0.181 | 0.516 ± 0.148 | 1.295 ± 0.218 | 1.265 ± 0.214 | 1.314 ± 0.211 | 1.195 ± 0.223 | 1.295 ± 0.214 | 1.675 ± 0.323 | 1.495 ± 0.324 | 1.323 ± 0.216 | 1.262 ± 0.344 | 1.260 ± 0.179 | 1.351 ± 0.196 | 1.046 ± 0.266 | 0.631 ± 0.179 | 2.200 ± 0.241 | 0.676 ± 0.178 | 0.880 ± 0.197 | 0.896 ± 0.213 | 0.838 ± 0.244 |
|  | $D_{H}$ (↓) | 0.259 ± 0.250 | 11.982 ± 8.459 | 13.080 ± 10.814 | 24.318 ± 14.282 | 33.946 ± 15.794 | 8.140 ± 8.376 | 30.604 ± 12.894 | 30.080 ± 18.217 | 35.596 ± 17.440 | 25.688 ± 12.263 | 9.213 ± 5.565 | 33.759 ± 14.568 | 37.355 ± 11.442 | 25.410 ± 14.487 | 29.930 ± 11.425 | 35.280 ± 5.150 | 28.276 ± 11.475 | 16.909 ± 9.314 | 31.071 ± 11.254 | 29.353 ± 9.834 | 30.991 ± 12.365 | 32.639 ± 9.157 |

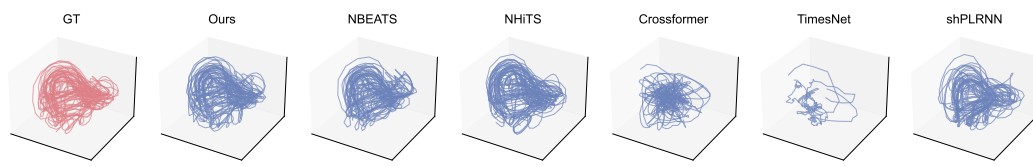

Figure 6: Visualization of forecasting results on Lorenz96 system. We reduce the dimensionality of the high-dimensional system to three dimensions using Principal Component Analysis (PCA) to facilitate this visualization.

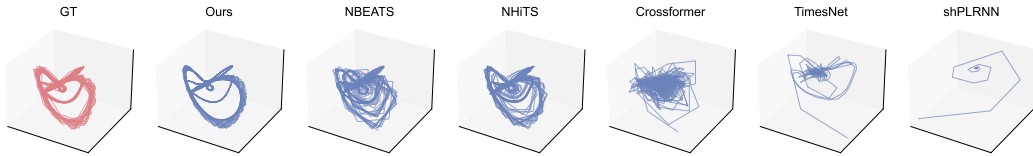

Figure 7: Visualization of forecasting results on ECG system. We reduce the dimensionality of the high-dimensional system to three dimensions using Principal Component Analysis (PCA) to facilitate this visualization.

### E.2 DETAILED RESULTS OF ABLATION STUDY

We present a comprehensive table of ablation study results, including 95% confidence intervals, for both the Rossler and EEG datasets in Table 5. Note that a version of this table without confidence interval information is provided in the main text. Additionally, ablation study results with 95% confidence intervals for the Lorenz63, Lorenz96, and ECG datasets are presented in Table 6. The results clearly demonstrate the effectiveness of each component in our model design across a range of systems, including systems of different dimensions. Removing any single component generally leads to a noticeable degradation in performance, indicating that each design choice contributes meaningfully to the overall model. Specifically, the full model consistently achieves the best or second-best scores across most metrics, including point-wise accuracy (e.g., 1-step Error, sMAPE) and measures of attractor geometry preservation (e.g., $D_{\text{frac}}$, $D_{\text{stsp}}$). This suggests that our design not only enhances prediction accuracy but also helps maintain the underlying structural and dynamic properties of the original systems. Overall, the ablation results validate that our architectural components play crucial and complementary roles in capturing both short-term dynamics and long-term attractor characteristics.

### E.3 DETAILED RESULTS OF MODEL ROBUSTNESS

**Robustness against Noise and Training Data Ratio.** We present detailed tables demonstrating our model's performance with different noise intensities in Table 7, including the results of 95% confidence intervals, on five simulated and real-world datasets. These results clearly demonstrate the robustness of our model under increasing levels of noise. Even at high noise intensities, the model maintains competitive accuracy and stable dynamical consistency across all datasets, as indicated by both the sMAPE@1 and $D_{\text{stsp}}$ metrics. This highlights the model's strong resilience to noisy inputs, a crucial property for real-world applications where data contamination is often unavoidable. Table 8 reveals that our model achieves high predictive accuracy even with minimal training data. Remarkably, with as little as 20% of the training data, the model already surpasses the fully-trained best baseline discussed in the main text. This data efficiency underscores the model's capacity to generalize from limited samples, making it especially valuable in domains where labeled data is scarce or expensive to obtain. For better visualization, we also illustrate the results of model robustness against observational noises and training data on Lorenz63 and Lorenz96 systems, and ECG dataset in Figure 10 and Figure 11, respectively.

**Robustness against Observation Length.** We investigate the impact of historical observation length on model performance during the inference stage. Owing to our model's generative training paradigm mainly based on next-patch prediction, the length of historical input can be flexibly adjusted at

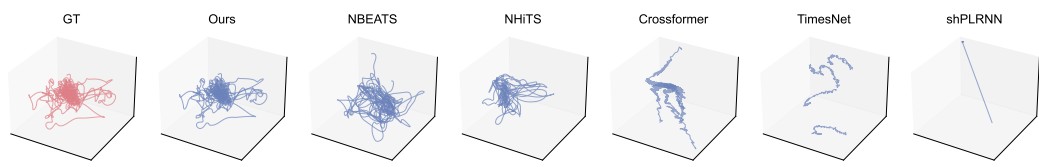

Figure 8: Visualization of forecasting results on EEG system. We reduce the dimensionality of the high-dimensional system to three dimensions using Principal Component Analysis (PCA) to facilitate this visualization.

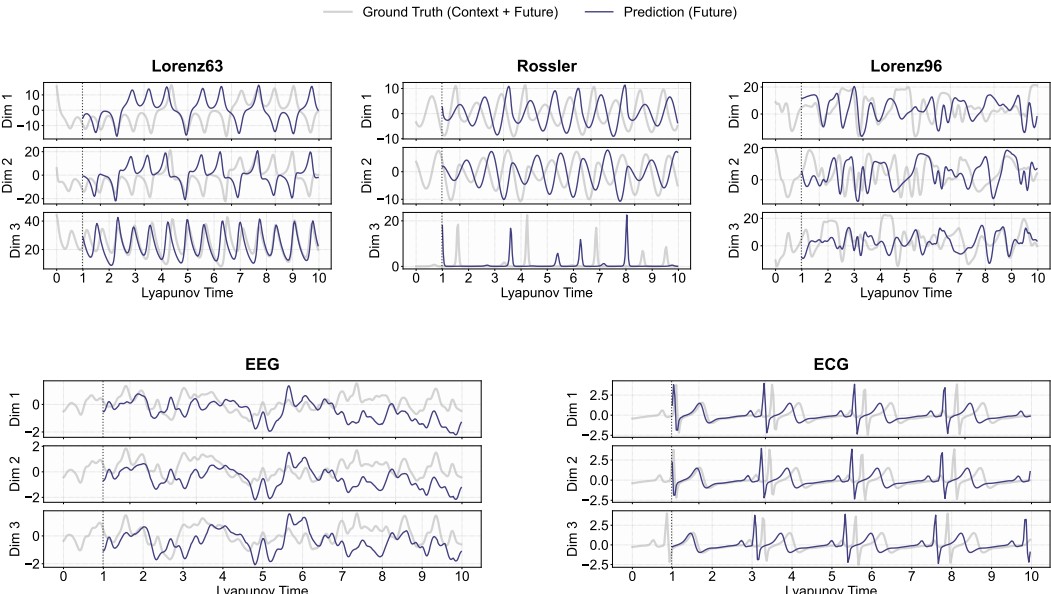

Figure 9: Dimension-wise comparison between our predictions and the ground truth for all systems.

inference time, enabling exploration of its influence on forecasting accuracy. Since our approach incorporates a patching and embedding process for system representation, the observation length must be chosen as an integer multiple of the patch size. For the specific patch sizes used across the five systems, please refer to Table 3. The results are shown in Figure 12, from which we draw two main conclusions:

- **First**, for the short-term prediction metric (sMAPE@1), performance generally degrades as the historical observation length decreases. This is expected, as chaotic systems are inherently sensitive to initial conditions. Insufficient historical context makes it difficult to accurately locate the current state of the system on the attractor, leading to less precise short-term forecasts. Nevertheless, our model demonstrates notable efficiency: for some systems, such as Lorenz96 and EEG, the model achieves better accuracy with only $0.25T_L$ and $0.2T_L$ of observation length, respectively, compared to the fully-trained baseline with $1T_L$ of observation length. This highlights the model's ability to leverage limited information for effective short-term prediction.

- **Second**, for the long-term metric ($D_{\text{stsp}}$), performance remains remarkably stable across different observation lengths. This reflects a key strength of our generative framework: even with minimal historical input and reduced point-wise accuracy, the model is still capable of producing trajectories whose underlying geometric structure closely resembles that of the true attractor. In fact, for the majority of systems, trajectories generated with just $0.2T_L$ of historical input exhibit greater geometric similarity to the ground truth than those produced by the fully-trained baseline with $1T_L$ of historical observation length. This suggests that the generative model effectively captures the global structure of the attractor, supporting robust long-term dynamics reconstruction even under limited observational constraints.

Table 5: Model performances for Rossler (3D) and EEG (64D) when removing each of our designs. Reported values represent the mean ± 95% confidence interval (CI), where CIs are computed based on multiple runs from different randomly sampled initial conditions to capture variability in the model's predictions. The best performance of each metric is marked in **bold**, and the second-best performance is underlined.

| System / Metric Ablation | Rossler (3D) | | | | | | EEG (64D) | | | | | |
|---|---|---|---|---|---|---|---|---|---|---|---|---|
| | VPT (↑) | 1-step Error (↓) | sMAPE@1 / 4 / 10 (↓) | | | $D_{frac}$ (↓) | $D_{stsp}$ (↓) | VPT (↑) | 1-step Error (↓) | sMAPE@1 / 4 / 10 (↓) | | | $D_{frac}$ (↓) | $D_{stsp}$ (↓) |
| Full | **9.83 ± 0.20** | **0.007 ± 0.001** | **2.84 ± 0.18** | **3.55 ± 0.17** | **12.64 ± 1.94** | **0.050 ± 0.008** | **0.034 ± 0.009** | **9.43 ± 1.24** | **0.031 ± 0.003** | **14.21 ± 2.06** | **18.12 ± 9.35** | **21.87 ± 16.93** | **0.180 ± 0.106** | **0.259 ± 0.530** |
| w/o PGR | 2.48±0.36 | 0.087±0.014 | 17.91±1.77 | 38.31±2.17 | 56.43±2.09 | 0.105±0.012 | 0.239±0.039 | 0.036±0.05 | 0.144±0.011 | 74.79±9.94 | 109.16±14.94 | 129.81±8.91 | 0.646±0.198 | 12.410±9.085 |
| w/o HA | 5.51±0.45 | 0.055±0.009 | 11.29±1.01 | 25.07±1.38 | 46.55±2.36 | 0.075±0.010 | 0.150±0.028 | 0.044±0.06 | 0.161±0.023 | 42.69±4.75 | 47.40±8.22 | 54.37±14.55 | **0.143±0.076** | 0.927±0.790 |
| w/o MTP | 5.66±0.65 | 0.063±0.012 | 12.48±1.21 | 22.99±1.57 | 41.47±2.93 | 0.091±0.012 | 0.193±0.039 | 0.045±0.06 | 0.114±0.011 | 48.13±7.06 | 57.18±9.55 | 78.87±19.75 | 0.424±0.168 | 2.220±1.028 |
| w/o SF | 1.29±0.41 | 0.071±0.008 | 27.22±0.83 | 37.18±1.47 | 55.86±2.43 | 0.108±0.012 | 0.230±0.041 | 2.31±1.72 | 0.041±0.003 | 24.34±3.36 | 33.17±9.64 | 42.87±17.14 | 0.278±0.144 | 0.263±0.168 |
| w/o MMD | 7.40±0.48 | 0.031±0.005 | 8.34±0.72 | 17.23±1.13 | 37.32±2.70 | 0.083±0.012 | 0.136±0.024 | 4.81±3.05 | 0.056±0.006 | 23.22±3.70 | 27.71±7.88 | 31.48±12.57 | 0.230±0.128 | 0.479±0.867 |
| Encoder-oriented | 2.61±0.49 | 0.054±0.010 | 20.68±1.47 | 35.14±2.71 | 56.12±2.90 | 0.081±0.011 | 0.160±0.031 | 0.017±0.04 | 0.083±0.021 | 120.90±9.30 | 121.31±7.69 | 150.46±6.61 | 0.749±0.106 | 23.629±9.520 |

Table 6: Model performances for Lorenz63 (3D), Lorenz96 (5D), and ECG (5D) when removing each of our designs. Reported values represent the mean ± 95% confidence interval (CI), where CIs are computed based on multiple runs from different randomly sampled initial conditions to capture variability in the model's predictions. The best performance of each metric is marked in **bold**, and the second-best performance is underlined.

| System / Metric Ablation | Lorenz63 (3D) | | | | | | Lorenz96 (5D) | | | | | | ECG (5D) | | | | | |
|---|---|---|---|---|---|---|---|---|---|---|---|---|---|---|---|---|---|---|
| | VPT (↑) | 1-step Error (↓) | sMAPE@1 / 4 / 10 (↓) | | | $D_{frac}$ (↓) | $D_{stsp}$ (↓) | VPT (↑) | 1-step Error (↓) | sMAPE@1 / 4 / 10 (↓) | | | $D_{frac}$ (↓) | $D_{stsp}$ (↓) | VPT (↑) | 1-step Error (↓) | sMAPE@1 / 4 / 10 (↓) | | | $D_{frac}$ (↓) | $D_{stsp}$ (↓) |
| Full | **5.06 ± 0.38** | **0.080 ± 0.011** | **3.26 ± 0.56** | **22.05 ± 3.13** | **27.29 ± 3.49** | **0.046±0.007** | **1.013±0.294** | **1.66 ± 0.19** | **0.135 ± 0.015** | **20.28 ± 2.69** | **81.18 ± 4.45** | **113.56 ± 2.37** | **0.154 ± 0.020** | 15.440 ± 0.799 | **0.93 ± 0.45** | **0.017 ± 0.006** | **37.14 ± 15.75** | **73.64 ± 11.33** | **99.68 ± 11.93** | **0.036 ± 0.013** | **0.004 ± 0.001** |
| w/o PGR | 3.74±0.33 | 0.138±0.013 | 7.00±1.36 | 38.25±4.32 | 75.73±2.82 | 0.046±0.007 | 1.013±0.294 | 1.64±0.16 | 0.085±0.012 | 21.53±2.26 | 82.99±4.16 | 113.66±2.22 | 0.136±0.018 | 15.747±0.739 | 0.92±0.47 | 0.018±0.005 | 41.02±16.19 | 75.87±10.65 | 100.05±11.75 | 0.043±0.013 | 0.006±0.002 |
| w/o HA | 4.62±0.36 | 0.098±0.015 | 4.73±1.26 | 24.46±3.50 | 71.81±3.17 | 0.045±0.006 | 1.160±0.289 | 1.64±0.17 | 0.125±0.016 | 22.40±2.77 | 82.05±4.47 | 113.38±2.20 | **0.125±0.017** | 15.180±0.761 | 0.80±0.35 | 0.026±0.007 | 43.05±15.54 | 75.34±10.81 | 100.90±11.45 | 0.044±0.012 | 0.011±0.003 |
| w/o MTP | 5.02±0.28 | 0.090±0.009 | 3.45±0.59 | 25.70±2.69 | 68.47±2.69 | 0.051±0.008 | 1.218±0.409 | 1.52±0.16 | 0.109±0.014 | 21.06±2.49 | 85.48±4.19 | 115.74±2.03 | 0.128±0.020 | **15.102±0.688** | 0.90±0.36 | 0.018±0.005 | 40.20±15.73 | 74.18±9.92 | 100.65±11.45 | 0.043±0.013 | 0.010±0.004 |
| w/o SF | 1.77±0.30 | 0.437±0.069 | 23.98±4.19 | 70.84±5.32 | 96.05±3.19 | 0.099±0.010 | 1.195±1.166 | 1.52±0.16 | 0.095±0.009 | 22.66±3.02 | 87.47±4.44 | 117.71±2.30 | 0.121±0.018 | 15.732±0.822 | 0.75±0.30 | 0.016±0.005 | 43.80±16.08 | 77.73±11.47 | 100.90±11.55 | 0.042±0.016 | 0.012±0.005 |
| w/o MMD | 4.54±0.37 | 0.167±0.027 | 4.78±0.69 | 23.98±3.43 | 70.67±3.31 | 0.049±0.007 | 1.199±0.318 | 1.33±0.14 | 0.169±0.020 | 24.14±2.53 | 86.12±3.92 | 115.75±1.83 | 0.153±0.022 | 15.288±0.700 | 0.60±0.32 | 0.036±0.011 | 49.29±18.19 | 89.41±6.18 | 109.14±7.21 | 0.269±0.045 | 0.099±0.027 |
| Encoder-oriented | 2.16±0.24 | 0.335±0.049 | 11.61±1.41 | 55.52±3.94 | 97.46±2.14 | 0.276±0.015 | 5.308±0.620 | 0.10±0.02 | 0.071±0.077 | 77.88±3.24 | 117.70±1.79 | 127.78±1.04 | 0.171±0.026 | 36.038±1.198 | 0.44±0.25 | 0.047±0.015 | 53.48±17.48 | 100.36±9.83 | 124.04±5.07 | 0.905±0.082 | 0.200±0.035 |

### E.4 HYPERPARAMETER STUDY

We investigate the impact of patch size, multi-patch prediction steps, and Mamba layers on the model performance, and illustrate the results in Figure 13, Figure 14, and Figure 15, respectively. We draw the following conclusions from the results:

**Impact of Patch Size.** Patch size plays a crucial role in determining representation quality, particularly in time series data where patches often capture local semantic patterns. An appropriately chosen patch size can enhance the model's ability to extract meaningful temporal features, whereas suboptimal sizing may either omit important short-term dynamics or dilute significant signals. Therefore, tailoring the patch size to the specific characteristics of the target system is essential for optimal performance.

**Impact of Multi-patch Prediction Steps.** The number of multi-patch prediction steps also affects the model's capacity to learn temporal dependencies. A step size that is too small limits the model's ability to capture the long-term dynamics necessary for accurate forecasting, while an excessively large step size may hinder next-patch prediction performance and increase computational complexity. Hence, it is imperative to strike a balance: the chosen multi-patch prediction steps should be sufficiently large to reflect system-level temporal evolution, yet not so large as to compromise local accuracy or computational efficiency.

**Impact of Mamba Layers.** The depth of Mamba layers—stacked in a residual manner—represents different hierarchical components of the underlying system dynamics. Each additional layer introduces a new level of abstraction, enabling the model to capture more complex temporal structures. However, deeper architectures also incur higher computational costs. Therefore, selecting an appropriate number of Mamba layers is key to achieving an effective trade-off between representational power and inference efficiency.

### E.5 COMPLEXITY ANALYSIS

We analyze model complexity by evaluating the number of parameters and the inference time required to forecast a trajectory of length $10T_L$. For physics-informed machine learning models, we include Koopa, PLRNN, PRC, and HoGRC as representative baselines. For general time series forecasting models, we consider NBEATS, NHiTS, CrossFormer, PatchTST, TimesNet, TiDE, and iTransformer. The results across five simulated and real-world datasets are presented in Table 9. From the results, we draw three conclusions:

- **First**, our model achieves a favorable trade-off between parameter count and inference time. Compared to the majority of general time series forecasting models, our model has significantly fewer

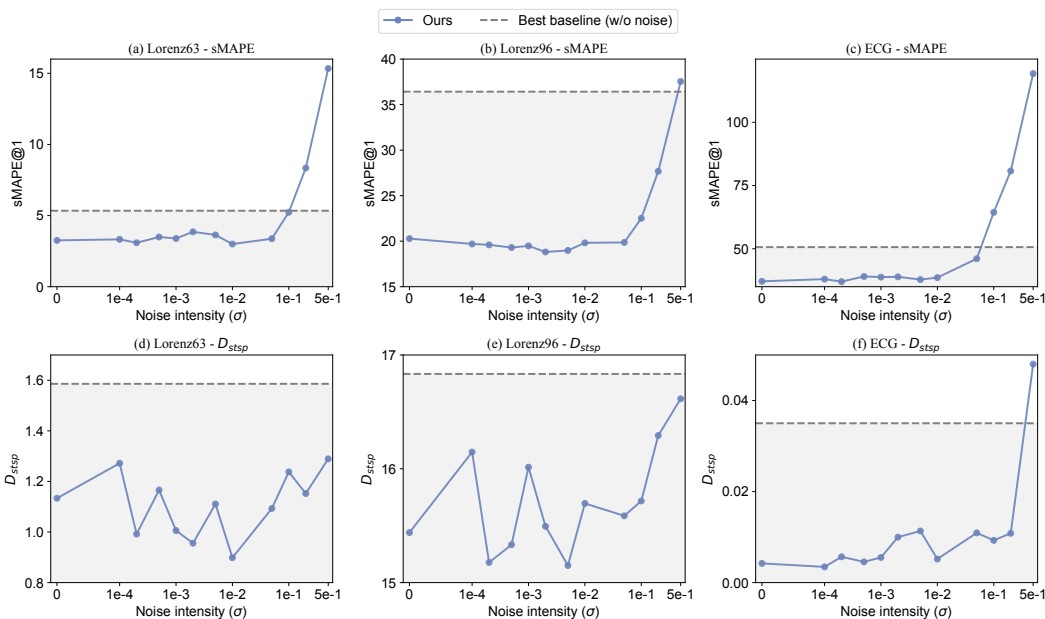

Figure 10: Model robustness against noise on Lorenz63 and Lorenz 96 systems, and ECG dataset.

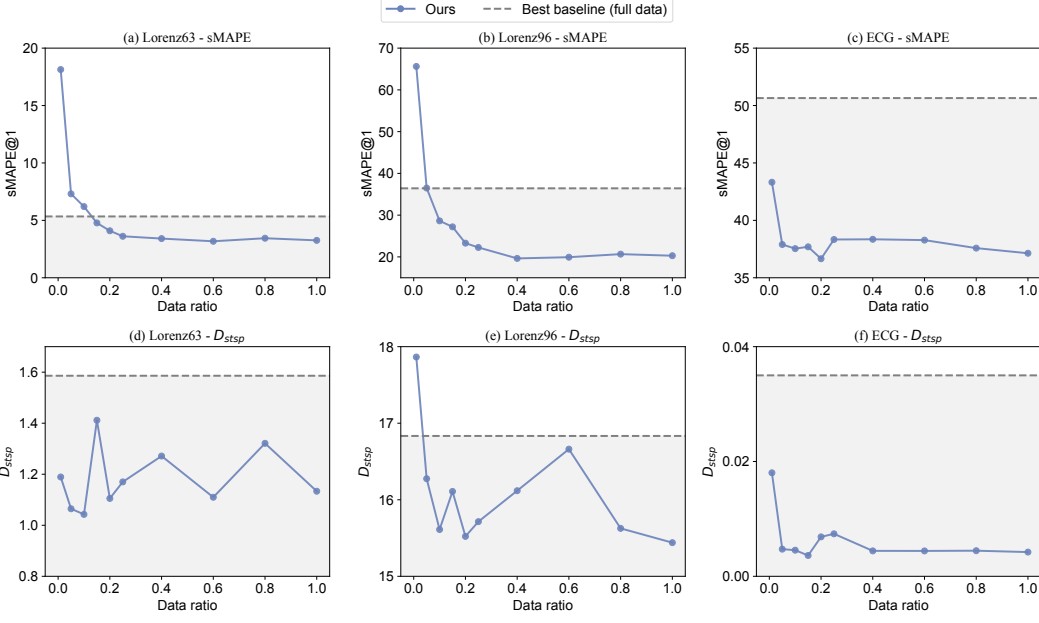

Figure 11: Model robustness against the number of training data on Lorenz63 and Lorenz96 systems, and ECG dataset.

parameters while delivering superior performance. Its inference time is moderate—substantially shorter than that of competitive baselines such as NBEATS and NHiTS. On the other hand, although our model has a larger number of parameters than physics-informed machine learning models—owing to its deep learning architecture—it benefits from shorter inference times, highlighting its efficiency during deployment.

• **Second**, inference time is closely related to the patch size used in our model. For example, on the ECG dataset, where a smaller patch size is employed (see Table 3), the model requires more computation to forecast a trajectory of the same length. However, as demonstrated above,

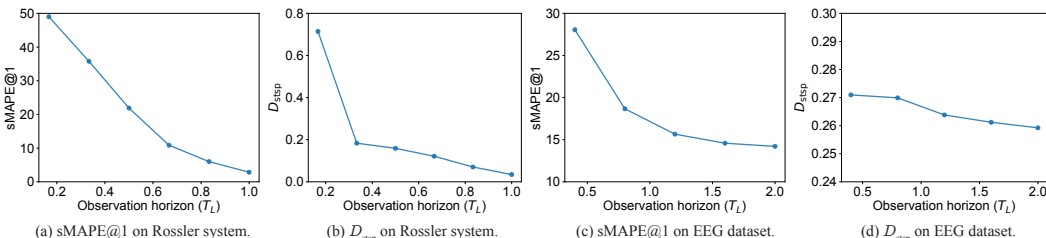

(a) sMAPE@1 on Rossler system.  (b) $D_{\text{stsp}}$ on Rossler system.  (c) sMAPE@1 on EEG dataset.  (d) $D_{\text{stsp}}$ on EEG dataset.

Figure 12: The impact of historical observation length on the model performance during the inference stage. Reported lines and shades represent the mean ± 95% confidence interval (CI), where CIs are computed based on multiple runs from different randomly sampled initial conditions to capture variability in the model's predictions.

Table 7: Model robustness against different noise intensities. Reported values represent the mean ± 95% confidence interval (CI), where CIs are computed based on multiple runs from different randomly sampled initial conditions to capture variability in the model's predictions.

| System | Lorenz63 (3D) | | Rossler (3D) | | Lorenz96 (5D) | | ECG (5D) | | EEG (64D) | |
|---|---|---|---|---|---|---|---|---|---|---|
| Metric / Intensity | sMAPE@1 (↓) | $D_{\text{stsp}}$ (↓) | sMAPE@1 (↓) | $D_{\text{stsp}}$ (↓) | sMAPE@1 (↓) | $D_{\text{stsp}}$ (↓) | sMAPE@1 (↓) | $D_{\text{stsp}}$ (↓) | sMAPE@1 (↓) | $D_{\text{stsp}}$ (↓) |
| 0.0000 | 3.26 ± 0.56 | 1.133 ± 0.340 | 2.85 ± 0.17 | 0.034 ± 0.009 | 20.28 ± 2.69 | 15.440 ± 0.799 | 37.14 ± 15.75 | 0.004 ± 0.001 | 14.21 ± 2.06 | 0.259 ± 0.530 |
| 0.0001 | 3.33 ± 0.72 | 1.272 ± 0.345 | 2.21 ± 0.22 | 0.049 ± 0.012 | 19.70 ± 2.49 | 16.147 ± 0.719 | 38.00 ± 13.40 | 0.003 ± 0.003 | 13.70 ± 1.98 | 0.338 ± 0.695 |
| 0.0002 | 3.09 ± 0.58 | 0.992 ± 0.246 | 3.49 ± 0.26 | 0.064 ± 0.016 | 19.60 ± 2.52 | 15.177 ± 0.670 | 37.01 ± 12.94 | 0.006 ± 0.002 | 20.90 ± 2.91 | 0.268 ± 0.477 |
| 0.0005 | 3.49 ± 0.83 | 1.166 ± 0.324 | 2.79 ± 0.27 | 0.054 ± 0.014 | 19.30 ± 2.49 | 15.333 ± 0.650 | 39.04 ± 14.21 | 0.005 ± 0.002 | 24.18 ± 3.40 | 0.413 ± 0.698 |
| 0.0010 | 3.39 ± 0.64 | 1.006 ± 0.301 | 3.45 ± 0.27 | 0.063 ± 0.017 | 19.48 ± 2.54 | 16.014 ± 0.841 | 38.80 ± 13.59 | 0.006 ± 0.002 | 25.81 ± 3.40 | 0.664 ± 1.210 |
| 0.0020 | 3.86 ± 0.88 | 0.956 ± 0.255 | 3.84 ± 0.26 | 0.057 ± 0.021 | 18.82 ± 2.47 | 15.495 ± 0.700 | 38.90 ± 13.42 | 0.010 ± 0.004 | 58.42 ± 7.09 | 4.978 ± 0.938 |
| 0.0050 | 3.64 ± 1.00 | 1.111 ± 0.253 | 5.43 ± 0.29 | 0.056 ± 0.017 | 18.98 ± 2.51 | 15.151 ± 0.663 | 37.83 ± 13.36 | 0.011 ± 0.003 | 64.81 ± 5.23 | 8.772 ± 0.885 |
| 0.0100 | 3.00 ± 0.64 | 0.898 ± 0.255 | 9.30 ± 0.43 | 0.067 ± 0.017 | 19.82 ± 2.44 | 15.696 ± 0.824 | 38.60 ± 13.40 | 0.005 ± 0.002 | 49.06 ± 3.58 | 2.044 ± 1.098 |
| 0.0500 | 3.37 ± 1.03 | 1.093 ± 0.261 | 27.97 ± 0.89 | 0.082 ± 0.029 | 19.86 ± 2.27 | 15.587 ± 0.742 | 46.06 ± 14.21 | 0.011 ± 0.002 | 73.08 ± 5.33 | 8.712 ± 1.363 |
| 0.1000 | 5.24 ± 1.35 | 1.238 ± 0.259 | 37.80 ± 1.05 | 0.095 ± 0.015 | 22.51 ± 2.56 | 15.717 ± 0.756 | 64.42 ± 17.91 | 0.009 ± 0.004 | 40.88 ± 4.63 | 0.614 ± 0.888 |
| 0.2000 | 8.34 ± 2.08 | 1.152 ± 0.248 | 44.63 ± 1.63 | 0.110 ± 0.018 | 27.67 ± 2.87 | 16.292 ± 0.915 | 80.75 ± 15.92 | 0.011 ± 0.003 | 53.63 ± 2.30 | 0.858 ± 1.126 |
| 0.5000 | 15.34 ± 2.11 | 1.289 ± 0.495 | 55.49 ± 1.72 | 0.182 ± 0.020 | 37.54 ± 3.24 | 16.615 ± 0.669 | 119.26 ± 9.06 | 0.048 ± 0.006 | 107.81 ± 3.28 | 14.910 ± 0.975 |

patch size has a significant impact on model performance. This necessitates a deliberate trade-off between computational efficiency and predictive accuracy, particularly when deploying the model in latency-sensitive scenarios.

- **Third**, general time series forecasting models typically involve a high parameter count and tend to incur longer inference times. In contrast, while physics-informed models often maintain a low parameter count, they usually perform short-step or single-step predictions (as seen with PRC and HoGRC), which results in longer cumulative inference time when forecasting long sequences. This highlights the limitation of such approaches in terms of scalability.

In summary, our model strikes a compelling balance between complexity and performance. It maintains acceptable levels of parameterization and time cost while delivering often superior forecasting accuracy, making it a well-rounded and scalable solution for diverse temporal prediction tasks.

Table 8: Model robustness against different training data ratios. Reported values represent the mean ± 95% confidence interval (CI), where CIs are computed based on multiple runs from different randomly sampled initial conditions to capture variability in the model's predictions.

| System | Lorenz63 (3D) | | Rossler (3D) | | Lorenz96 (5D) | | ECG (5D) | | EEG (64D) | |
|---|---|---|---|---|---|---|---|---|---|---|
| Metric
Ratio | sMAPE@1 ($\downarrow$) | $D_{\text{stsp}}$ ($\downarrow$) | sMAPE@1 ($\downarrow$) | $D_{\text{stsp}}$ ($\downarrow$) | sMAPE@1 ($\downarrow$) | $D_{\text{stsp}}$ ($\downarrow$) | sMAPE@1 ($\downarrow$) | $D_{\text{stsp}}$ ($\downarrow$) | sMAPE@1 ($\downarrow$) | $D_{\text{stsp}}$ ($\downarrow$) |
| 0.01 | 18.13 ± 2.66 | 1.189 ± 0.197 | 20.62 ± 0.99 | 0.212 ± 0.037 | 65.62 ± 4.53 | 17.865 ± 0.711 | 43.32 ± 16.31 | 0.018 ± 0.005 | 127.12 ± 16.44 | 13.779 ± 6.439 |
| 0.05 | 7.31 ± 1.12 | 1.065 ± 0.332 | 16.91 ± 0.95 | 0.204 ± 0.037 | 36.49 ± 4.15 | 16.275 ± 0.762 | 37.90 ± 10.55 | 0.005 ± 0.002 | 104.38 ± 15.50 | 9.823 ± 6.068 |
| 0.10 | 6.20 ± 0.94 | 1.043 ± 0.239 | 13.22 ± 0.98 | 0.183 ± 0.037 | 28.63 ± 3.70 | 15.611 ± 0.748 | 37.54 ± 11.80 | 0.005 ± 0.003 | 99.27 ± 10.55 | 12.505 ± 9.182 |
| 0.15 | 4.78 ± 0.81 | 1.412 ± 0.523 | 10.16 ± 0.84 | 0.173 ± 0.035 | 27.19 ± 3.58 | 16.109 ± 0.801 | 37.70 ± 10.75 | 0.004 ± 0.002 | 85.34 ± 13.22 | 13.470 ± 9.944 |
| 0.20 | 4.09 ± 0.61 | 1.105 ± 0.236 | 9.85 ± 0.57 | 0.109 ± 0.017 | 23.29 ± 3.11 | 15.521 ± 0.676 | 36.66 ± 11.05 | 0.007 ± 0.003 | 94.21 ± 13.40 | 10.335 ± 7.027 |
| 0.25 | 3.61 ± 0.58 | 1.170 ± 0.316 | 10.72 ± 0.91 | 0.168 ± 0.034 | 22.26 ± 2.76 | 15.714 ± 0.711 | 38.33 ± 11.56 | 0.007 ± 0.003 | 71.56 ± 11.16 | 12.066 ± 9.347 |
| 0.40 | 3.41 ± 0.59 | 1.271 ± 0.438 | 5.23 ± 0.47 | 0.092 ± 0.020 | 19.64 ± 2.42 | 16.118 ± 0.710 | 38.35 ± 12.55 | 0.004 ± 0.002 | 27.31 ± 3.83 | 0.473 ± 0.722 |
| 0.60 | 3.18 ± 0.56 | 1.110 ± 0.266 | 4.93 ± 0.34 | 0.072 ± 0.014 | 19.93 ± 2.32 | 16.661 ± 0.778 | 38.28 ± 13.07 | 0.004 ± 0.002 | 20.26 ± 2.94 | 0.630 ± 1.189 |
| 0.80 | 3.44 ± 0.76 | 1.321 ± 0.530 | 5.12 ± 0.49 | 0.085 ± 0.015 | 20.66 ± 2.53 | 15.626 ± 0.834 | 37.58 ± 13.50 | 0.004 ± 0.002 | 18.36 ± 3.92 | 0.455 ± 1.366 |
| 1.00 | 3.26 ± 0.56 | 1.133 ± 0.340 | 2.85 ± 0.17 | 0.034 ± 0.009 | 20.28 ± 2.69 | 15.440 ± 0.799 | 37.14 ± 15.75 | 0.004 ± 0.001 | 14.21 ± 2.06 | 0.259 ± 0.530 |

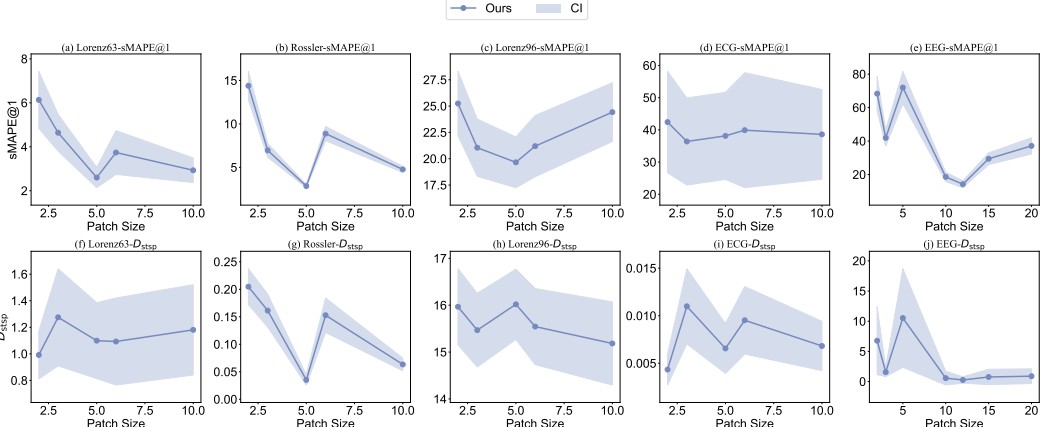

Figure 13: The impact of patch size on the model performance. Reported lines and shades represent the mean ± 95% confidence interval (CI), where CIs are computed based on multiple runs from different randomly sampled initial conditions to capture variability in the model's predictions.

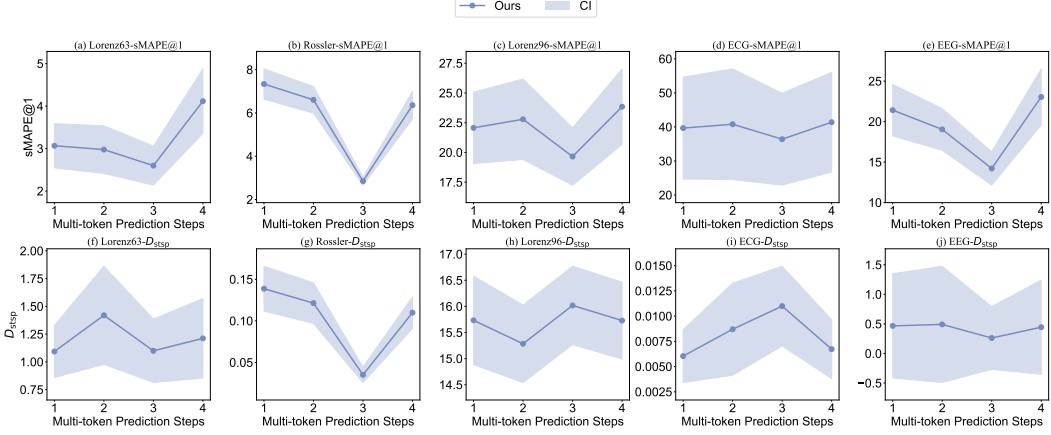

Figure 14: The impact of multi-patch prediction steps on the model performance. Reported lines and shades represent the mean ± 95% confidence interval (CI), where CIs are computed based on multiple runs from different randomly sampled initial conditions to capture variability in the model's predictions.

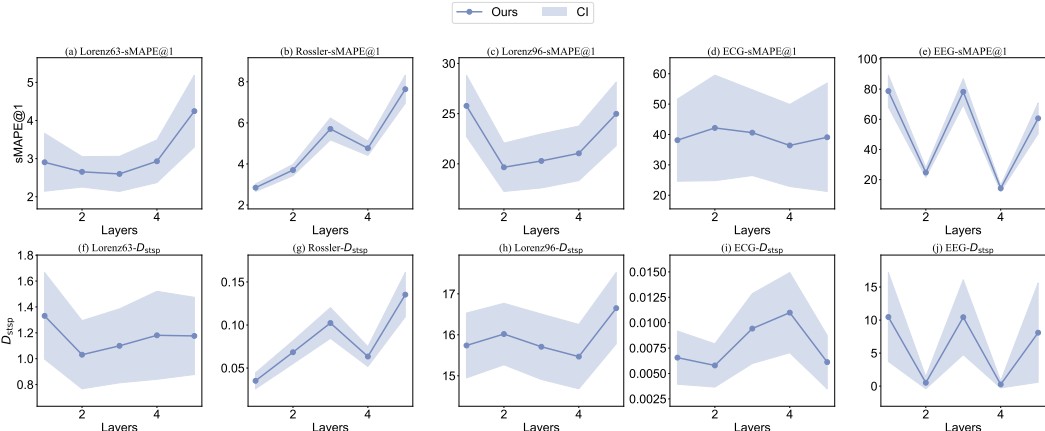

Figure 15: The impact of Mamba layers on the model performance. Reported lines and shades represent the mean ± 95% confidence interval (CI), where CIs are computed based on multiple runs from different randomly sampled initial conditions to capture variability in the model's predictions.

Table 9: Model complexity analysis: parameter counts and inference time (mean ± standard deviation) across different simulated and real-world datasets.

| Method | Lorenz63 (3D) | | Rossler (3D) | | Lorenz96 (5D) | | ECG (5D) | | EEG (64D) | |
|---|---|---|---|---|---|---|---|---|---|---|
| | Param (M) | Time (ms) | Param (M) | Time (ms) | Param (M) | Time (ms) | Param (M) | Time (ms) | Param (M) | Time (ms) |
| Ours | 3.65 | 183.76 ± 2.38 | 0.97 | 106.12 ± 3.42 | 2.76 | 285.19 ± 2.67 | 3.63 | 935.39 ± 28.51 | 4.56 | 321.49 ± 15.90 |
| NBEATS | 6.24 | 2052.07 ± 285.74 | 6.24 | 2418.02 ± 830.10 | 6.24 | 2051.68 ± 118.35 | 6.24 | 2183.45 ± 130.93 | 6.24 | 4098.94 ± 113.18 |
| NHiTS | 7.17 | 1599.60 ± 263.96 | 7.17 | 1637.66 ± 326.75 | 7.17 | 1625.34 ± 303.85 | 7.17 | 1694.81 ± 39.67 | 7.17 | 3200.06 ± 116.81 |
| CrossFormer | 7.56 | 176.62 ± 13.28 | 7.56 | 174.77 ± 12.27 | 7.57 | 173.80 ± 12.16 | 7.57 | 180.66 ± 13.18 | 7.73 | 186.10 ± 12.39 |
| PatchTST | 6.04 | 33.13 ± 1.37 | 6.04 | 32.98 ± 1.15 | 6.04 | 33.25 ± 1.16 | 6.07 | 35.74 ± 1.33 | 6.23 | 36.70 ± 1.41 |
| TimesNet | 143.01 | 1105.63 ± 6.52 | 143.01 | 1118.16 ± 4.80 | 143.02 | 1118.32 ± 3.22 | 143.02 | 1180.06 ± 7.66 | 143.08 | 1415.14 ± 17.31 |
| TiDE | 0.60 | 57.12 ± 0.97 | 0.60 | 57.50 ± 0.77 | 0.63 | 86.80 ± 1.12 | 0.66 | 86.29 ± 0.95 | 2.56 | 921.20 ± 4.98 |
| iTransformer | 4.54 | 23.93 ± 0.38 | 4.54 | 24.17 ± 0.87 | 4.54 | 39.64 ± 0.90 | 4.55 | 23.98 ± 0.68 | 4.57 | 23.87 ± 0.50 |
| Koopa | 0.09 | 51.21 ± 1.22 | 0.09 | 51.31 ± 1.06 | 0.10 | 51.07 ± 0.93 | 0.10 | 51.24 ± 1.76 | 0.56 | 50.68 ± 0.86 |
| PLRNN | 0.37 | 855.34 ± 35.56 | 0.37 | 845.33 ± 20.60 | 0.43 | 929.98 ± 32.50 | 0.12 | 2283.17 ± 88.50 | 0.11 | 10453.53 ± 780.31 |
| PRC | 0.003 | 2160.28 ± 57.71 | 0.003 | 2081.44 ± 40.71 | 0.003 | 3596.25 ± 55.42 | 0.003 | 793.72 ± 5.43 | 0.003 | 3112.54 ± 20.65 |
| HoGRC | 0.003 | 1775.32 ± 10.23 | 0.003 | 1911.88 ± 5.86 | 0.003 | 2772.48 ± 40.89 | 0.003 | 594.21 ± 6.92 | 0.003 | 26455.86 ± 3056.85 |

