# OpenReview forum: "Mamba Integrated with Physics Principles Masters Long-term Chaotic System Forecasting"
_ICLR.cc/2026/Conference — Submitted to ICLR 2026_

### Official Review · Reviewer_K6po · 2025-10-19

**Soundness:** 1
**Presentation:** 3
**Contribution:** 2
**Rating:** 4
**Confidence:** 5

**Summary:**

The authors propose a Mamba-based architecture for predicting chaotic systems. By incorporating physics-based inductive biases into their model, they show it can be effective in capturing both the short-term and long-term dynamics of chaotic systems.

**Strengths:**

Forecasting chaotic systems is a challenging and important problem in many domains. Adapting general sequence modeling architectures such as Mamba to chaotic systems is interesting and might lead to new directions in scientific machine learning.

**Weaknesses:**

* The baselines are incomplete. In particular, the authors did not compare with foundation models designed for dynamical systems, such as Panda (Lai et al. 2025) and DynaMix (Hemmer and Durstewitz 2025). It may also be interesting to compare with recently proposed naive baselines such as context parroting (Zhang and Gilpin 2025).
* The datasets for the benchmark only contain five systems, three of them low-dimensional toy systems, which is inadequate. To demonstrate the robustness of the conclusions, please include more chaotic systems with diverse characteristics in the benchmark.

**Questions:**

* The problem of long-horizon forecasting based on short context trajectory is an inherently ill-defined problem. I am skeptical that any model can perform well consistently across all chaotic systems (no free lunch theorems). Can authors elaborate on why they think their model can be the exception?
* For example, within 1 Lyapunov Time, the trajectory may not even explore both butterfly wings of the Lorenz63 attractor. How can PhyxMamba capture the long-term dynamics and attractor geometry in this case?
* For the results in Table 1, I find it hard to believe that PhyxMamba can achieve a VPT of over 9 Lyapunov Times based on only 60 data points. Can authors provide more evidence supporting this extraordinary claim?
* Have the authors tested forecasting partially-observed chaotic systems (e.g., univariate forecasts)? The delay embedding used in PhyxMamba could be well suited to this challenging task.

---

### Official Review · Reviewer_fZEw · 2025-10-27

**Soundness:** 3
**Presentation:** 3
**Contribution:** 2
**Rating:** 2
**Confidence:** 4

**Summary:**

This study introduces PhyxMamba, a hybrid forecasting framework that combines a Mamba-based state-space model with physics-informed principles to generate long-term predictions of chaotic dynamical systems based on short-term observations. The proposed method takes low dimensional time series, and then reconstructs the system’s attractor manifold through time-delay embedding, creating a high-dimensional representation to recover global dynamical features. It then applies a generative training strategy with multi-patch prediction and a residual Mamba architecture to model long-range dependencies while maintaining physical coherence. To mitigate error accumulation and preserve the geometry of the strange attractor, PhyxMamba incorporates a student-forcing stage and Maximum Mean Discrepancy (MMD)–based regularization, in order to enforce statistical consistency between predicted and true trajectories. Experiments on simulated chaotic systems (Lorenz63, Rössler, Lorenz96) and real-world datasets (ECG, EEG) show that PhyxMamba achieves strong forecasting accuracy and recovers the invariant measures and geometric integrity of chaotic attractors. Ablation and robustness analyses show that the physics-guided embedding and generative design contribute to stable long-horizon prediction.

**Strengths:**

The Maximum Mean Discrepancy (MMD) regularizer is a nice approach. While this has previously been applied to time series datasets, it is not a widely-known method in this setting, and it appears to improve the author’s model’s performance on this dataset.

The residual stacking approach in the Mamba blocks is a great idea. This approach is inspired by the NBEATS papers, and it seems to help in this case.

I found the paper clear, well-structured, and easy-to-follow. The related work section covers the appropriate areas well. While I was not familiar with all terminology in the authors’ subfield, I found that the description was clear enough that I could follow the narrative.

The experiments are careful and the baselines are appropriate: multiple metrics are used to assess the quality of the forecasts over both long- and short- term intervals, and the authors ablate many of their custom architecture’s components to determine how much they contribute to the model.

**Weaknesses:**

1. The architecture seems ad hoc, like a collection of methods combined together, and I do not feel that it inherits the simplicity or theoretical advantages of Mamba2 and other models like it. The methods section describes a very complicated architecture and multistage training loop. The authors first perform multi-patch next step prediction with teacher forcing. They then perform student forcing, checking that the model’s own predictions capture long-term structure. During the latter loop, the authors enforce regularization terms that ensure history-prediction consistency, as well as prediction-ground-truth consistency. All of these choices introduce substantial complexity into the training loop, including hyperparameters like the number of time delays, the patch size, embedding dimension, mamba depth, multi-patch depth, and the size of the various regularization terms.

2. Several architectural and methods choices do not fully make sense to me. The authors provide ablations showing that both time delay embedding and patching improve the model. But that doesn’t seem possible, since the two featurizations are redundant:  Since w are going to flatten into an embedding of shape N x (D*V*m), the patch size duplicates the time delays and vice versa. Additionally, my understanding is that Mamba is an SSM architecture that modifies an earlier time series SSM (S4) to work with language data. But the authors are now using Mamba for time series. This seems like an unusual choice, compared to using the S-series models, an RNN with spectral initialization, or a reservoir computer.

3. While I followed the time series modeling claims, I do not follow the chaotic dynamical systems results. The authors are learning to model a dynamical system based on a time series of length T, and they then aim to forecast that system for H >> T. But this does not seem possible. If I were fitting a linear dynamical system on T datapoints in D dimensions, the error of the fit would scale as 1/sqrt{T}. Wouldn’t these small errors exponentially blow up, precluding attaining the correct long-term statistics when H >> T?

4. Several strong and conceptually well-motivated domain-specific forecast architectures have been proposed for chaotic systems over the past few years, including next-generation reservoir computers and teacher-forced methods. While it is good that this particular model performs well on benchmarks, the problem area is narrow enough that I would prefer to see more conceptual novelty. Why wouldn’t a motivated practitioner with an unknown time series use TimeNet, a general model that performs well on general time series benchmarks like GIFT-Eval? Particularly because most time series we encounter in the wild have unknown provenance and thus unknown dynamical structure like stationarity. While not all models have to be intended for practical applications, I would like to see stronger conceptual insights, rather than the current complicated collection of small tweaks to existing architectures.

**Questions:**

*[Question 1]* I am confused by the finding that “Strong short-term predictive accuracy does not necessarily guarantee robust long-term forecasting performance.” Since the systems that the authors study are generated by ordinary differential equations, wouldn’t the best possible model approximate numerical integration to high precision? How can a model get long-term predictions right without mastering short term first?

*[Question 2]* What does a forecast for “1 Lyapunov time” mean in this context? If I understand Eq. 2 directly, this can be calculated directly from the ODE, but I don’t see how I would get a single number out, even for the “Lorenz” example equations that the authors provide. Do you find this empirically by analyzing your forecasts post-hoc? How does this work for the EEG and other time series?

*[Question 3]* How do you calculate the KL divergence between the forecasted time series and the ground truth? Since the output of PhyxMamba is a set of discrete timepoints, are you binning the output space in order to generate an empirical distribution? Does that bias the estimator in higher dimensions?

---

### Official Review · Reviewer_GCDL · 2025-10-30

**Soundness:** 2
**Presentation:** 2
**Contribution:** 2
**Rating:** 4
**Confidence:** 4

**Summary:**

The paper introduces PhyxMamba, a Mamba-based state-space modeling with physics-informed principles for long-term chaotic system forecasting from short historical data. It reconstructs attractor manifolds via time-delay embeddings, employs generative multi-patch training, and preserves attractor geometry through distribution regularization. Experiments on simulated and real-world datasets (Lorenz, Rössler, ECG, EEG) demonstrate superior accuracy and stability over state-of-the-art models in both short- and long-horizon predictions, while maintaining key dynamical and statistical properties of chaotic systems.

**Strengths:**

The paper has the following strengths worth credits:

1. Both simulation and real benchmarks are evaluated to demonstrate the generalisation of PhyxMamba on chaos systems;
2. Building physics constrained Mamba model for chaos systems learning is a novel attempt.
3. Introducing the multi-patch prediction induced learning constraint is a novel attempt.

**Weaknesses:**

The paper should be improved considering the following facts:

1. The paper evaluates PhyxMamba on simulated ODE chaos systems, not covering simulated PDE chaos systems and more high-dimensional cases, which is a significant overclaim of PhyxMamba as a framework for chaotic systems.
2. Motivations are not clear. The necesscity of using short history input compared with 1-step input in the literature is not specified.  To which concern that predicting autoregressively with Mamba is important/advanced compared with other methods, such as Markov Neural Operators and constrained Koopman Operators, is not analysed or discussed.
3. Limited theoretical analysis of convergence and stability is provided for the student-forcing patch decomposition method and its optimzation.
4. Given the strength of mamba models in interpretability on physical systems analogy, the paper lacks essential exploration and demonstation of hidden states in PhyxMamba, how the components of hidden states are learnt and aligned with the analogy.
5. Data in Table 4 is hard to read.

**Questions:**

1. Given the setting in 3.3, the paper focus on researching methods for dissipative chaotic systems. The paper only shows the trajectory results on the invariant measure. Can the paper elaborate on how the method handles the transition from dissipation to  the invariant measure both theoratically and experimentally?
2. The paper demonstrated PhyxMamba's performance on relatively low dimensional on chaos systems, and the simulations are mainly ode based. How PhyxMamba perform on more complexed PDE chaos systems?
3. Regarding the computation complexity of MMD, can the paper elaborate on how this term would work and scale on training PhyxMamba for high dimensional chaotic systems, e.g 2D/3D turbulence?
4. Regarding the short history input, the paper claims using a window of steps covering one Lyapunov time for forecasting. Is it a necessity that  at leaset 'one Lyapunov time' long window is used in every inference? What would the model behave if a shorter length / an incomplete Lyapunov time window is given in inference?
5. Following on question 4, what would the model behave when the input window has a coarse time steps? (e.g. instead of having 30 points in a Lyapunov time window, only 3-5 points covering a Lyapunov time window are sampled to make the dataset)

Happy to think about increasing my score if the paper is significantly improved to address the questions and weakness concerns.

---

### Meta-Review · Area_Chair_dM1y · 2025-12-15

**Summary:**

The paper introduces PhyxMamba, designed for long-term forecasting of chaotic systems. The method uses a Mamba-based state-space model, manifold reconstruction using time-delayed embeddings, and a generative training scheme with multi-patch prediction with attractor geometry regularization based on maximum mean discrepancy. The approach is evaluated on simulated systems and real-world datasets.

While the integration of Mamba with physics-informed constrained is conceptually interesting, the reviewers identified technical concerns to which the authors did not provide rebuttals during the discussion period. Examples of such include the lack of benchmarking and baselines in more challenging systems, and implausibility of predicting multiple Lyapunov times into the future (suggesting potential overfitting or issues with the evaluation protocol).

Given the severity of outstanding technical concerns, the recommendation is rejection.

**Reviewer Concerns:**

No rebuttals provided.

**Reviewer Scores:**

Scores would not have changed because authors did not provide rebuttals.

---

### Decision · Program_Chairs · 2026-01-26

Reject